# Peak refreezing in the Greenland firn layer under future warming scenarios

**Brice Noël** [1] ✉, **Jan T. M. Lenaerts** [2], **William H. Lipscomb**[3], **Katherine Thayer-Calder**[3] & **Michiel R. van den Broeke** [1]

Firn (compressed snow) covers approximately 90% of the Greenland ice sheet (GrIS) and currently retains about half of rain and meltwater through refreezing, reducing runoff and subsequent mass loss. The loss of firn could mark a tipping point for sustained GrIS mass loss, since decades to centuries of cold summers would be required to rebuild the firn buffer. Here we estimate the warming required for GrIS firn to reach peak refreezing, using 51 climate simulations statistically downscaled to 1 km resolution, that project the long-term firn layer evolution under multiple emission scenarios (1850–2300). We predict that refreezing stabilises under low warming scenarios, whereas under extreme warming, refreezing could peak and permanently decline starting in southwest Greenland by 2100, and further expanding GrIS-wide in the early 22$^{nd}$ century. After passing this peak, the GrIS contribution to global sea level rise would increase over twenty-fold compared to the last three decades.

Firn, the perennial layer of compressed snow, covers about 90% of the Greenland ice sheet (GrIS). Liquid water percolates into the firn pore space where it can be retained by capillary forces[1], refrozen as semi-impermeable ice lenses[2], or stored in perennial firn aquifers[3,4]. The firn layer currently retains ~45% of liquid water from rain and surface meltwater, or ~230 Gt yr$^{-1}$ (Gigaton or km$^3$ per year; 1960–2014)[5]. These retention mechanisms thus prevent a significant fraction of rain and meltwater from running off, mitigating surface mass loss. Moreover, firn is significantly brighter than the underlying bare ice, which further limits GrIS melt and runoff. Following recent Arctic warming, surface melt and subsequent runoff have increased quadratically with temperature[6], accelerating GrIS mass loss[7,8] and significantly contributing to global sea level rise[9]. Runoff acceleration is partly due to firn retreat exposing dark bare ice in expanding ablation zones[10,11], combined with reduced firn retention capacity through the formation of thick, impermeable ice lenses[12], progressively depleting pore space. At the same time, atmospheric warming favours rainfall[13,14] at the expense of snowfall, limiting the recovery of the firn layer[15]. The loss of firn could mark a tipping point for GrIS mass loss as decades to centuries of cold summers would be required to rebuild the porous firn layer. In a warming climate, we expect the refreezing of liquid water in

the firn to reach a maximum before it irreversibly declines. The timing of this 'peak refreezing' is relevant for projections of future GrIS mass loss and the central topic of this paper.

In a previous study, we showed that Greenland peripheral ice caps have already reached peak refreezing in the late 1990s[16]. Passing this peak initiated a permanent refreezing decline that still continues today, tripling mass loss rates. Likewise, reduced refreezing has accelerated the mass loss of Canadian[17] and Svalbard glaciers[18]. In a future warming climate, it is likely that the much larger and higher GrIS will also reach peak refreezing and progressively lose its firn buffer. Since firn currently retains about half of surface melt and rainfall, passing the refreezing peak would at least double runoff and subsequent surface mass loss. Here we investigate the long-term (up to 2300) resilience of the GrIS firn layer by estimating the atmospheric warming required to reach peak refreezing. We further explore how passing this threshold would affect the future GrIS contribution to global sea level rise.

## Results

### Climate models and scenario projections

We use climate projections from the Community Earth System Model (CESM2) at 0.9 × 1.25° spatial resolution[19] forced by low-end and

[1]Institute for Marine and Atmospheric research Utrecht, Utrecht University, Utrecht, Netherlands. [2]Department of Atmospheric and Oceanic Sciences, University of Colorado Boulder, Boulder, CO, USA. [3]Climate and Global Dynamics Laboratory, National Center for Atmospheric Research, Boulder, CO, USA. ✉e-mail: b.p.y.noel@uu.nl

high-end Shared Socioeconomic Pathway emission scenarios (SSP1-2.6 and SSP5-8.5)[20] covering the period 1950–2099. These projections are dynamically downscaled with a regional climate model (RACMO2.3p2)[21] to 11 km, and further statistically downscaled to 1 km resolution (see "Methods")[22]. For model evaluation, we compare both CESM2-forced RACMO2.3p2 projections at 1 km with a benchmark RACMO2.3p2 simulation (1958–2020)[10] forced by a combination of ERA-40[23], ERA-Interim[24] and ERA5 climate reanalyses[25], that is further statistically downscaled to 1 km resolution.

To sample other warming trajectories, we complement the two RACMO2.3p2 projections above with CESM2 outputs from 11 pre-industrial (1850–1949) and 12 historical simulations (1950–2014), extended by 25 scenario projections (2015–2100) including from lowest to highest emission: SSP1-2.6 (6 members), SSP2-4.5 (6 members), SSP3-7.0 (5 members) and SSP5-8.5 (8 members). Among these projections, one SSP1-2.6 and two SSP5-8.5 members extend to 2300; one of the SSP5-8.5 simulations includes an interactively coupled ice sheet, i.e., it accounts for ice dynamics including glacier retreat and thinning (see "Methods"). As intermediate scenarios are not defined beyond 2100 in O'Neill et al. (2015)[20], CESM2 projections under SSP2-4.5 and SSP3-7.0 were not extended to 2300. All these

CESM2 simulations are directly statistically downscaled from the native model resolution to the 1 km grid (see "Methods"). The resulting set of 51 downscaled products reconstructs 450 years (1850–2300) of GrIS surface mass balance (SMB), i.e., the difference between the mass gained from snow accumulation, and the mass lost from meltwater runoff and sublimation, at 1 km resolution.

## Greenland firn under low and high-end warming by 2100

The mass balance (SMB minus solid ice discharge) of the GrIS remained close to zero[8], or was slightly negative[7], until the early 1990s, with firn covering the accumulation zone that represents 92 ± 2% of the ice sheet area (Fig. 1a). Our CESM2-forced RACMO2.3p2 historical simulation (1950–2014) reproduces an ice sheet in approximate mass balance before the 1990s with surface mass gain (SMB = 430 Gt yr$^{-1}$ in 1950–1990; Fig. 1c, d)[21] that almost compensates for dynamic mass loss from solid ice discharge across the grounding line of marine-terminating glaciers (∼460 Gt yr$^{-1}$)[7]. From the 1990s onward, the GrIS started to lose mass at an accelerating rate[8]. This is captured by our historical product, i.e., −9.4 ± 1.6 Gt yr$^{-2}$, in excellent agreement with −9.4 ± 1.2 Gt yr$^{-2}$ measured by the Gravity Recovery and Climate Experiment (GRACE) for the period 2003–2014[21]. Our historical

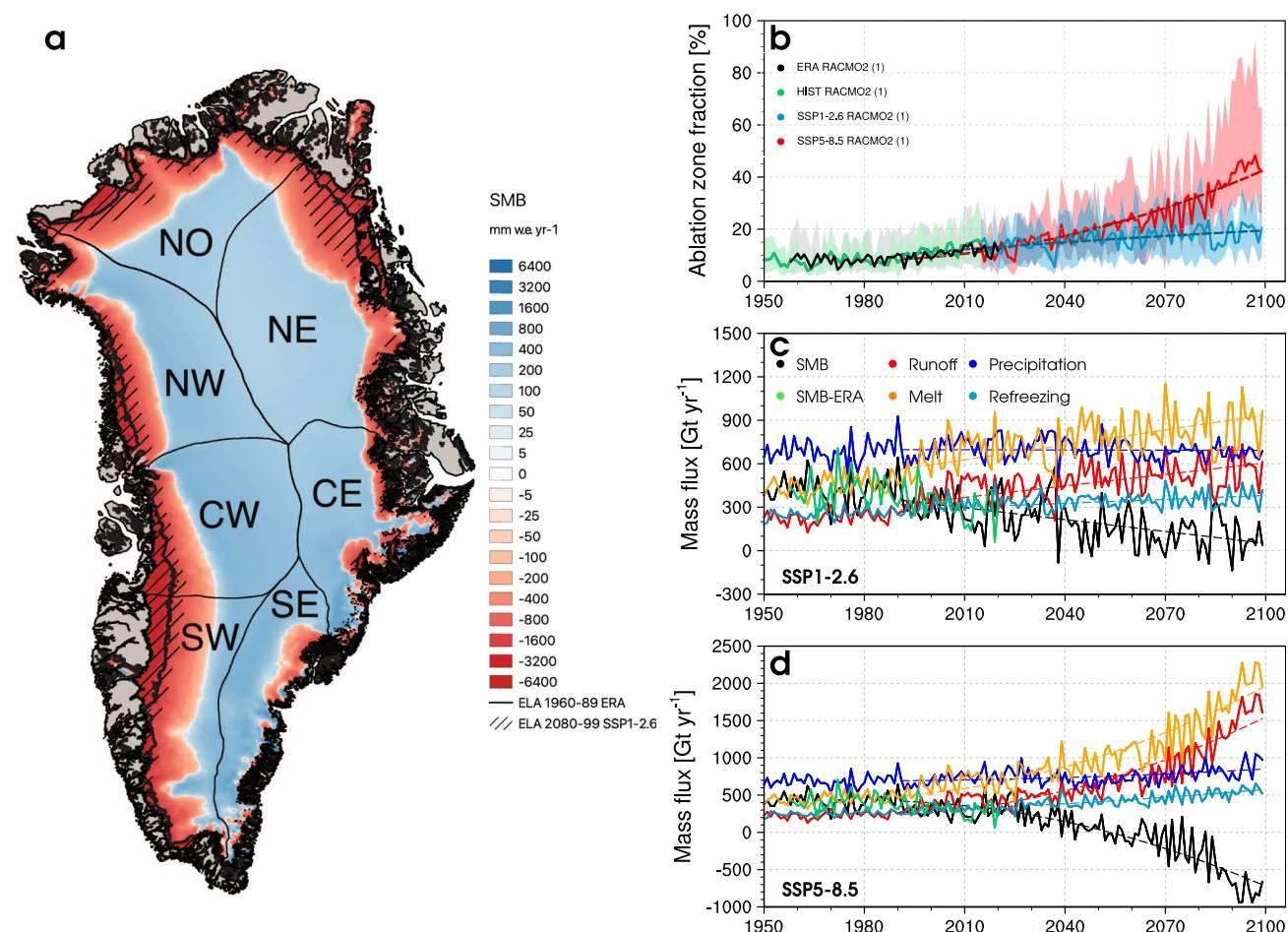

**Fig. 1 | GrIS SMB under low and high-end warming by 2100. a** Downscaled GrIS surface mass balance (SMB) from CESM2-forced RACMO2.3p2 at 1 km under SSP5-8.5 in 2080–2099. Black contour and hatching outline the equilibrium line altitude (ELA; SMB = 0) for the historical period 1950–1990 and for SSP1-2.6 in 2080–2099 respectively. The GrIS is divided in seven sectors[7] including north (NO), northwest (NW), northeast (NE), central west (CW), central east (CE), southwest (SW) and southeast (SE). **b** Time series of GrIS area fraction situated in the ablation zone (SMB < 0) from CESM2-forced RACMO2.3p2 for the historical period (1950–2014),

SSP1-2.6 and SSP5-8.5 (2015–2099) scenarios. The coloured bands in **b** span the minimum and maximum ablation zone fraction of individual GrIS sectors. **c** and **d** Time series of GrIS integrated SMB and components including runoff, total precipitation including snowfall and rainfall, melt and refreezing for the period 1950–2099 under SSP1-2.6 and SSP5-8.5 respectively. Linear and quadratic fits are represented as coloured dashed curves. The black line in **b** and green lines in **c**, **d** represent the RACMO2.3p2 simulation forced by ERA reanalyses (1958–2020)[10].

product also shows good agreement with the benchmark reanalysis-forced RACMO2.3p2 present-day simulation (black line in Fig. 1b and green line in Fig. 1c, d). Detailed evaluations of historical SMB (1950–2014) in native CESM2 and CESM2-forced RACMO2.3p2, i.e., including in situ measurements, remote sensing records and previous modelling efforts, are presented in Van Kampenhout et al.[26] and Noël et al.[21]. Overall, CESM2-forced RACMO2.3p2 shows very good agreement with SMB observations, with only a small bias in both the accumulation (−21 mm w.e.) and ablation zone (180 mm w.e.), of similar quality to the benchmark reanalysis-forced RACMO2.3p2 product (−22 mm w.e. and 120 mm w.e., respectively)[21]. This agreement is confirmed for GrIS-integrated SMB components (Supplementary Fig. 7), and for SMB on the regional scale (Supplementary Fig. 8).

We extend the historical SMB time series (1950–2014) with two projections for a low-end and high-end emission scenario (2015–2099)[27]. To explore regional changes in firn processes, we divide the GrIS in seven sectors[7] (Fig. 1a): southwest (SW), southeast (SE), central west (CW), central east (CE), north (NO), northwest (NW) and northeast (NE). Under the low-end warming trajectory (SSP1-2.6), we predict that the firn zone will remain extensive, covering 82% of the GrIS area in 2080–2099, with a minimum of 71% in SW Greenland (Fig. 1a). Following a linear expansion of the ablation zone area (Fig. 1b), SMB declines linearly (−2.3 Gt yr$^{-2}$ for 1991–2099) as a result of a linear increase in meltwater runoff (+1.9 Gt yr$^{-2}$) combined with an

insignificant change in total precipitation (−0.4 Gt yr$^{-2}$, $P > 0.01$) (Fig. 1c). The increase in runoff is primarily driven by enhanced melt (+2.5 Gt yr$^{-2}$) that is partly refrozen in the firn zone (+0.6 Gt yr$^{-2}$) (Fig. 1c). In contrast, we predict that the firn zone will rapidly retreat inland under the high-end warming scenario (SSP5-8.5), covering only 61% of the GrIS area in 2080–2099, with a minimum of 33% in SW Greenland (Fig. 1a). For this scenario, the ablation zone expands quadratically (Fig. 1b) and triggers a rapid SMB decline (−9.7 Gt yr$^{-2}$ for 1991–2099), the result of increased melt (+12.7 Gt yr$^{-2}$) that far exceeds the small increase in total precipitation (+1.3 Gt yr$^{-2}$) ascribed to both a significant rainfall increase (+1.1 Gt yr$^{-2}$) with an insignificant snowfall change (+0.2 Gt yr$^{-2}$, $P > 0.01$). This additional liquid water flux from melt and rain is partly retained through enhanced refreezing (+2.7 Gt yr$^{-2}$), while the remainder is discharged to the ocean by increased runoff (+11.1 Gt yr$^{-2}$). Under both warming scenarios, firn pore space is progressively depleted as refreezing increases in a retreating firn zone. This mechanism reduces the firn refreezing capacity, defined here as the fraction of rain and meltwater retained or refrozen in firn (see "Methods"). Figure 2a highlights the loss of firn refreezing capacity, which drops from 53% in the period 1950–1990 down to 40% and 28% in 2080–2099 under the SSP1-2.6 and SSP5-8.5 scenarios. Individual sectors of Greenland follow a similar trend, with a minimum refreezing capacity of 31% and 19% in SW Greenland under SSP1-2.6 and SSP5-8.5, respectively.

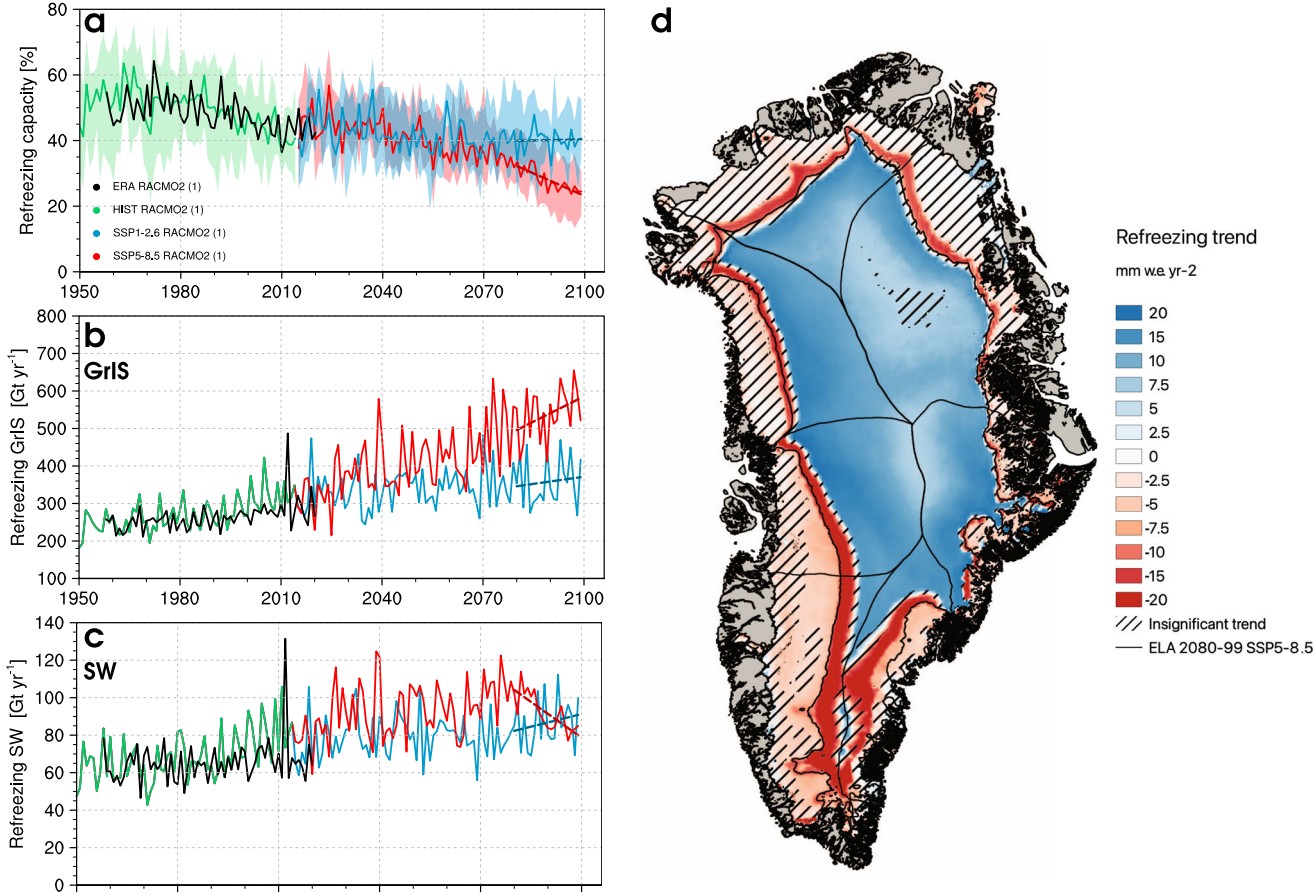

**Fig. 2 | GrIS refreezing trends under low and high-end warming by 2100. a** Time series of GrIS refreezing capacity from CESM2-forced RACMO2.3p2 for the historical period (1950–2014), SSP1-2.6 and SSP5-8.5 (2015–2099) scenarios. The coloured bands in **a** span the minimum and maximum refreezing capacity of individual GrIS sectors. **b** and **c** Time series of refreezing spatially integrated over the GrIS and SW sector for the historical, SSP1-2.6 and SSP5-8.5 simulations. The black line in **a–c** represents the reanalysis-forced RACMO2.3p2 simulation (1958–2020)[10]. **d** Refreezing trend from the CESM2-forced RACMO2.3p2 projection under SSP5-8.5 over 2080–2099. Black contour represents the equilibrium line altitude (ELA) in 2080–2099 and hatching outlines insignificant trends ($P > 0.05$). Dashed coloured lines in **a–c** represent trends for the period 2080–2099.

## Peak refreezing reached in south Greenland before 2100

Integrated over the GrIS, refreezing increases in time irrespective of the scenario (Fig. 2b). Figure 2d maps the GrIS-wide refreezing trend for the period 2080–2099 under SSP5-8.5. While refreezing significantly increases in the porous firn layer that remains in the interior far above the equilibrium line (SMB = 0), it slowly decreases below the equilibrium line as seasonal snowfall, which maintains small springtime refreezing, is gradually replaced by rainfall. In contrast, refreezing significantly declines in the low accumulation (percolation) zone, i.e., fringing the equilibrium line, as enhanced meltwater production progressively depletes the remaining firn pore space.

In contrast to the full GrIS, refreezing peaks and rapidly declines in SW Greenland from the 2080s onward under SSP5-8.5 ($-1.3$ Gt yr$^{-2}$ for 2080–2099) (Fig. 2c). This negative trend affects most of the sector area (Fig. 2d) that lies in the ablation zone (Fig. 1a). The negative trend results from a combination of reduced refreezing ($-1.3$ Gt yr$^{-2}$) and enhanced rainfall ($+1.4$ Gt yr$^{-2}$) that makes runoff ($+11.3$ Gt yr$^{-2}$) increase 30% faster than melt ($+8.6$ Gt yr$^{-2}$). In the absence of a snowfall increase ($-0.2$ Gt yr$^{-2}$, $P > 0.01$) to restore the firn buffer, we interpret this regional peak refreezing as a tipping point, accelerating the mass loss in SW Greenland. After passing this regional threshold under SSP5-8.5, the surface mass loss acceleration almost doubles ($+9.8$ Gt yr$^{-2}$, 2080–2099) compared to present-day ($+5.1$ Gt yr$^{-2}$ for 1991–2020). In SE Greenland, refreezing in 2080–2099 shows a small but insignificant negative trend ($-0.3$ Gt yr$^{-2}$; $P > 0.01$) under SSP5-8.5 (Supplementary Fig. 1g). This suggests that peak refreezing is also reached around 2100 in SE Greenland. Sectors in central and north Greenland show a persistent positive refreezing trend in 2080–2099 for both scenarios (Supplementary Fig. 1a–f). Consequently, refreezing still increases GrIS-wide by 2100 under SSP5-8.5 (Fig. 2b).

## Predicted regional and GrIS-wide peak refreezing beyond 2100

To identify whether other sectors and the GrIS as a whole will reach peak refreezing, we complement our two CESM2-forced RACMO2.3p2 projections with 48 downscaled CESM2 simulations (see "Methods"). These simulations include 25 CESM2 ensemble projections until 2100 (SSP1-2.6, SSP2-4.5, SSP3-7.0 and SSP5-8.5), as well as three long-term CESM2 projections until 2300 under extended SSP1-2.6 and SSP5-8.5 scenarios (see "Methods"). To estimate the regional timing of peak refreezing, even for projections that do not extend beyond 2100, we extrapolate the runoff line altitude in time, defined as demarcating the region producing over 100 mm w.e. of runoff per year (see "Methods"). To do this, we use as a benchmark the results from the SW sector where peak refreezing is reached before 2100 in all SSP5-8.5 scenarios (8 members) (Fig. 3d). We find that as firn saturates and retreats in SW Greenland, the runoff line migrates upward in a non-linear fashion under SSP5-8.5 (Supplementary Fig. 3a), from 1537 ± 228 m a.s.l. (metres above sea level) in 1960–1989 to a threshold altitude of 2514 ± 159 m a.s.l., when peak refreezing is crossed (Supplementary Fig. 2a). Peak refreezing in SW is reached when 90 ± 10% of the sector area produces runoff, i.e., 90% of the sector area is situated below the runoff line. Applying this 90 ± 10% condition to sector hypsometries, i.e., area-elevation distribution, yields regional thresholds for runoff line altitude (blue lines in Supplementary Fig. 2). The timing of peak refreezing in each sector is then obtained by extrapolating the increasing runoff line altitude until it crosses this threshold value (blue lines in Supplementary Fig. 3). Figure 3a–g show that the timing of peak refreezing in SSP5-8.5 obtained from extrapolation (red dots) agrees within uncertainty (red whiskers) with the peak identified in the two long-term refreezing projections under SSP5-8.5 (red lines), confirming the robustness of our method.

The results show that no other sector will pass peak refreezing by 2100 under any scenario (Fig. 3h). In 2080–2099, runoff is produced over sector areal fractions ranging from 39% in CW Greenland to 66% in the SE sector (Supplementary Fig. 2a–g and Fig. 3h). We estimate

that all sectors will reach the 90% runoff threshold in the 22$^{nd}$ century under SSP5-8.5 (Supplementary Fig. 3a–g), starting with SW Greenland (year 2089 ± 11) and followed by SE (2105 ± 11), NO (2118 ± 16), NE (2127 ± 11), NW (2136 ± 10), CE (2137 ± 8) and finally CW Greenland (2143 ± 9) (Fig. 3a–g and Supplementary Table 1). Likewise, we find that runoff is produced over 90% of the GrIS area when the runoff line reaches an average elevation of 2842 ± 276 m a.s.l. (blue line in Supplementary Fig. 2h). However, this elevation is not reached by 2100 in any of the warming scenarios (solid blue line in Fig. 4a–d). For instance, 52% of the GrIS area produces runoff in 2080–2099 under SSP5-8.5 (Fig. 3h). We predict that the GrIS will cross peak refreezing in year 2126 ± 14 under SSP5-8.5 (Fig. 5a, Supplementary Fig. 3h and Supplementary Table 1).

## Discussion

Three parameters control the differences in timing of regional peak refreezing: (i) the reference runoff line altitude when the GrIS was in approximate equilibrium in 1960–1989 (Supplementary Table 1), (ii) the threshold runoff line altitude to cross peak refreezing (Supplementary Table 1), and (iii) the rate of upward migration of the runoff line (Supplementary Fig. 3) that depends on the sector hypsometry (Supplementary Fig. 2) and the rate of regional atmospheric warming. The sectors in the south (SW and SE) pass peak refreezing first: they experience the warmest atmospheric conditions, and have the highest reference and the lowest threshold runoff line altitude. The sectors in the north (NO, NE and NW) have relatively gently sloping ice sheet margins and experience the fastest atmospheric warming in Greenland[10], resulting in the fastest upward migration of the runoff line (Supplementary Fig. 3c, d, g). NO crosses peak refreezing before NE and NW due to its lower elevated interior, and thus lower threshold altitude (Supplementary Fig. 2d). Central Greenland (CE and CW) passes peak refreezing the latest as it encompasses the highest regions of the GrIS, resulting in higher threshold runoff line altitudes (Supplementary Fig. 2), while experiencing slower upward migration of the runoff line (Supplementary Fig. 3b, f), equivalent to that of the south (Supplementary Fig. 3a, e).

We predict that for the GrIS peak refreezing is reached in year 2126 ± 14 under SSP5-8.5 (Fig. 5a). Figure 5b translates this threshold into an upper-atmospheric (500 hPa) global warming of 9.1 ± 0.9 °C relative to pre-industrial (1850–1949). As atmospheric warming continues beyond 2126, the firn refreezing capacity pursues its declining trend (Fig. 5c) until it stabilises at ~5% in the 2220s. Thereafter refreezing exclusively occurs in the shallow seasonal snowpack, buffering ~250 Gt yr$^{-1}$ of melt and rain in early summer (Fig. 5a). We identify similar residual refreezing for individual sectors of the GrIS, contributing approximately 15 Gt yr$^{-1}$ in SW, 80 Gt yr$^{-1}$ in NW, and about 30 Gt yr$^{-1}$ in other sectors (Fig. 3a–g). Under intermediate warming scenarios (SSP3-7.0), we extrapolate that peak refreezing could be reached about half a century later in year 2172 ± 20 (orange dot and whiskers in Fig. 5a). In contrast, under low-end warming scenarios (SSP1-2.6), upper-atmospheric temperature stabilises after 2100 (Fig. 5b), and firn refreezing capacity for the GrIS (Fig. 5a, c) and individual sectors (Fig. 3a–g) reach a plateau close to the 2080–2099 level. We anticipate similar results for the SSP2-4.5 scenarios, for which atmospheric temperature and firn refreezing stabilise in the period 2080–2099 (Fig. 5a, b). We conclude that strong (SSP1-2.6) and moderate mitigation scenarios (SSP2-4.5) prevent the occurrence of GrIS-wide peak refreezing.

Our estimates of the timing of (regional) peak refreezing should be considered an upper bound since additional processes not represented in our projections could further accelerate firn saturation and retreat. Among others, missing processes include the formation of thick, impermeable ice lenses inhibiting active meltwater percolation and retention in firn[12], although Langen et al.[28] previously showed that accounting for these in a regional climate model had little impact on

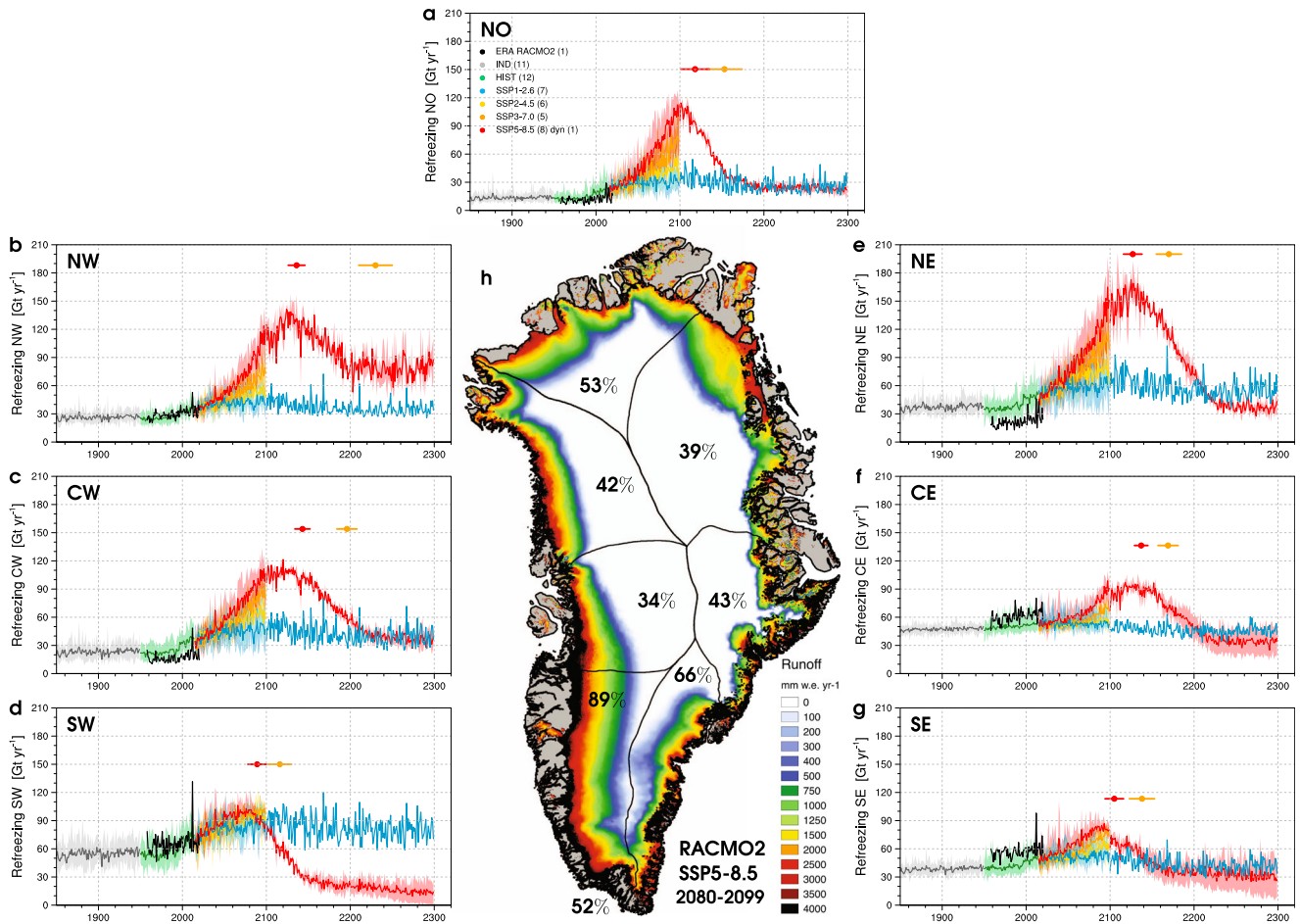

**Fig. 3 | Regional peak refreezing by 2200. a–g** Time series of sector-integrated refreezing for the period 1850–2300. Thick coloured lines represent ensemble mean refreezing at 1 km from 11 pre-industrial (IND), 12 historical (HIST) and 27 scenario projections, i.e., SSP1-2.6 (6 CESM2 and 1 RACMO2 members), SSP2-4.5 (6 members), SSP3-7.0 (5 members), and SSP5-8.5 (8 CESM2 and 1 RACMO2 members). One CESM2 SSP5-8.5 scenario includes ice dynamics (dyn). Coloured bands represent the maximum and minimum refreezing within each ensemble. Coloured dots and whiskers estimate the timing and uncertainties of the regional refreezing peak for SSP5-8.5 (red) and SSP3-7.0 (orange) using extrapolation of the runoff line altitude in time (see "Methods"). The black line in **a–g** represents the reanalysis-forced RACMO2.3p2 simulation (1958–2020)[10]. **h** Downscaled GrIS runoff from CESM2-forced RACMO2.3p2 at 1 km under SSP5-8.5 in 2080–2099. Numbers represent the land-ice area fraction producing significant runoff (>100 mm w.e. yr⁻¹) for the GrIS (bottom left) and individual sectors.

GrIS-integrated runoff. Furthermore, none of the models used here explicitly simulate fully saturated firn aquifers, although RACMO2.3p2 reliably captures the occurrence of perennial liquid water by capillary retention in the contemporary firn[3]. This means that present-day runoff from the active firn aquifers of SE Greenland is likely overestimated, potentially affecting the future evolution of the aquifers and their contribution to mass loss. Another missing process is the recent trend in more frequent summer atmospheric blocking over Greenland that amplifies melt since the early 2000s[29]. In a fully coupled run, i.e., considering dynamical adjustment of the ice sheet, glacier dynamics does not affect the timing of the refreezing peak and our conclusions, as runoff (Fig. 4e, f) and refreezing do not diverge before 2200 (Figs. 3a–g and 5a). In the 23$^{rd}$ century, rapid glacier thinning and retreat trigger a strong melt-elevation feedback that further reduces retention in the seasonal snowpack. In climate models with lower climate sensitivity[30], we expect that the dates of reaching the refreezing peak could be delayed but would occur at a similar level of warming.

In the absence of a significant snowfall increase under SSP5-8.5, passing peak refreezing can be interpreted as a tipping point, accelerating surface mass loss of the GrIS. Figure 5d shows that GrIS surface mass loss increases non-linearly with the upward migration of the runoff line, irrespective of the warming scenario. GrIS refreezing peaks when the runoff line reaches an average elevation of 2842 ± 276 m a.s.l. (Fig. 5e) and runoff is produced over 90% of the GrIS area. After passing this threshold, surface mass loss increased almost 5-fold relative to 2080–2099 (3334 ± 2276 Gt yr⁻¹; blue line in Fig. 5e) and raises global sea level by 9.2 ± 6.3 mm yr⁻¹, a 23-fold increase compared to 1992-2018 (0.4 mm yr⁻¹)[8]. A 9.1 ± 0.9 °C upper-atmosphere warming is required for GrIS refreezing to peak, demonstrating that only under the most extreme warming scenarios will peak refreezing be reached. This highlights the resilience of Greenland firn and its refreezing mechanism for the current ice sheet geometry.

## Methods
### Community Earth System Model (CESM2)
We use outputs from the CMIP6 Community Earth System Model version 2.1 (CESM2.1)[19] for 11 pre-industrial simulations (IND; 1850–1949), 12 historical simulations (HIST; 1950–2014), and 25 climate projections (SSP1-2.6 to SSP5-8.5) including 22 short-term (2015–2099) and 3 long-term members (2015–2300): one SSP1-2.6 and two SSP5-8.5. These simulations were produced and made available by the National Center for Atmospheric Research (NCAR). The different emission scenarios are described in O'Neill et al. (2015)[20], and account for increased radiative forcing by 2.6 W m⁻² (SSP1-2.6), 4.5 W

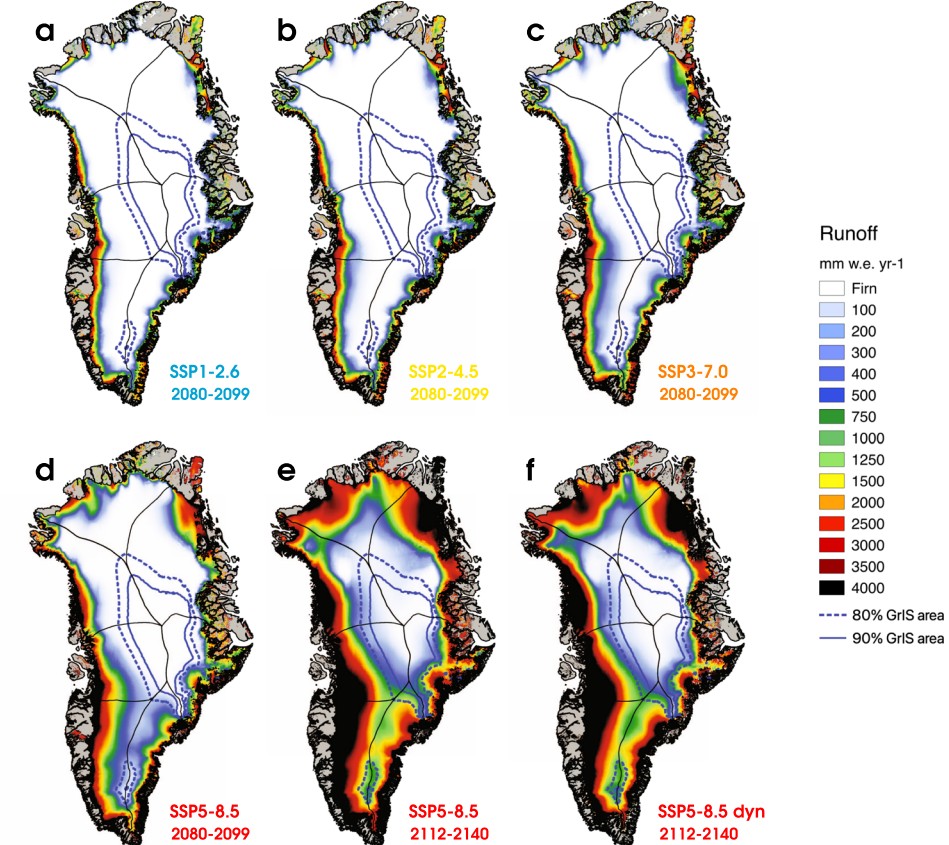

**Fig. 4 | GrIS runoff under various warming scenarios.** Ensemble-mean downscaled GrIS runoff at 1 km for the period 2080–2099 under **a** SSP1-2.6 (6 CESM2 and 1 RACMO2 members), **b** SSP2-4.5 (6 members), **c** SSP3-7.0 (5 members) and **d** SSP5-8.5 (8 CESM2 and 1 RACMO2 members). Downscaled GrIS runoff at 1 km for the period 2112-2140 (i.e., 2126 ± 14) from two long-term SSP5-8.5 projections: **e** excluding and **f** including ice dynamics (dyn). The solid and dashed blue contours represent the GrIS threshold runoff line altitude (90% of the area) and its associated lower bound uncertainty (80% of the area).

m$^{-2}$ (SSP2-4.5), 7.0 W m$^{-2}$ (SSP3-7.0) and 8.5 W m$^{-2}$ (SSP5-8.5) in 2100. The difference between each member of a same scenario stems from a random perturbation applied to the temperature field used for initialisation. CESM2 is an Earth System Model that simulates interactions between atmosphere-land-ocean systems on the global scale. The model includes the Community Atmosphere Model version 6 (CAM6)[31], the Parallel Ocean Program model version 2.1 (POP2.1)[32], the Los Alamos National Laboratory Sea Ice Model version 5.1 (CICE5.1)[33], the Community Land Model version 5 (CLM5)[34], and the Community Ice Sheet Model version 2.1 (CISM2.1)[35] to simulate interactions between atmosphere-ocean-land and sea ice dynamics as well as snow/ice surface processes in a fully coupled fashion. Here all components are active except for land-ice dynamics, i.e., excluding calving processes and subsequent ice sheet thinning and retreat. One SSP5-8.5 long-term projection uses the fully coupled CESM2-CISM2 to account for ice dynamics beyond 2100. In the latter product, CISM2 is first spun up over 9000 years using repeated CESM2 pre-industrial climate. Then CESM2-CISM2 is run interactively using a historical reconstruction (1950–2014) branched with a SSP5-8.5 projection (2015–2099) before modelling ice dynamics and geometry change in the period 2100–2300. Holding the ice sheet fixed until 2100 is justified as the geometry of the GrIS is not expected to change significantly by the end of the 21st century[36]. In CESM2, CAM6 and CLM5 run at a spatial resolution of 0.9 × 1.25° (~111 km) and exclusively prescribe land use changes, atmospheric greenhouse gas and aerosol emissions derived from the selected SSP warming scenarios[30]. CISM2 computes ice dynamics on a 4 km grid and, in fully coupled CESM2-CISM2 runs, returns the evolving GrIS geometry to CLM5. CLM5 has been adapted to simulate runoff, melt, firn retention and refreezing in a 10 m w.e.

snowpack[26], which does not account for impermeable ice lenses. CESM2 has been extensively used and evaluated to realistically represent contemporary[26] and projected GrIS SMB and components[36], despite its known higher climate sensitivity compared to CMIP5 predecessors[37].

**Regional Atmospheric Climate Model (RACMO2)**
We use the Regional Atmospheric Climate Model version 2.3p2 (RACMO2.3p2)[38] to dynamically downscale one historical CESM2 simulation (1950−2014) and two CESM2 projections including a low-end SSP1-2.6 and a high-end SSP5-8.5 warming scenario (2015−2099) to 11 km spatial resolution. The model incorporates the dynamical core of the High Resolution Limited Area Model (HIRLAM)[39] and the physics package cycle CY33r1 of the European Centre for Medium-Range Weather Forecasts-Integrated Forecast System (ECMWF-IFS)[40]. The polar (p) version of RACMO2 is specifically adapted to represent surface processes in polar regions, and includes a 40-layer snow module (up to 100 m depth) simulating melt, percolation and retention into firn and subsequent surface runoff[41]. Impermeable ice lenses are not represented in the snow module. The model represents dry-snow densification[42], drifting snow erosion[43], and snow albedo based on grain size, cloud optical thickness, solar zenith angle, and impurity content (soot)[44].

RACMO2 is forced by CESM2 climate outputs within a 24-grid-cell-wide relaxation zone at the lateral model boundaries. Forcing consists of temperature, pressure, specific humidity, wind speed and direction being prescribed at the 40 model atmospheric levels every 6 hours. Upper-atmospheric relaxation is active[45]. Sea ice extent and sea surface temperature are also prescribed on a

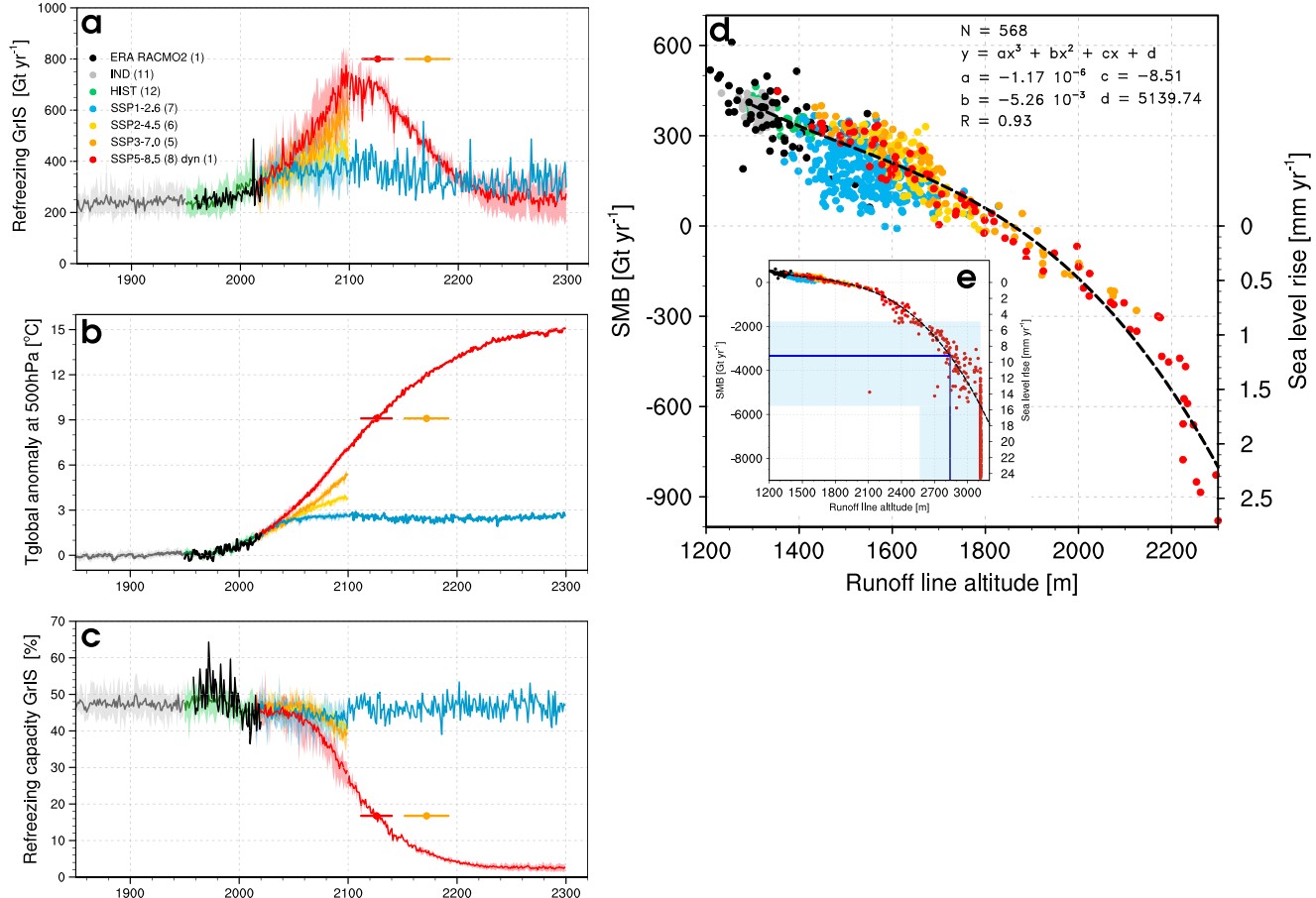

**Fig. 5 | GrIS peak refreezing and contribution to sea level rise. a** Time series of GrIS-integrated refreezing for the period 1850–2300. Thick coloured lines represent ensemble mean refreezing at 1 km from 11 pre-industrial (IND), 12 historical (HIST) and 27 scenario projections, i.e., SSP1-2.6 (6 CESM2 and 1 RACMO2 members), SSP2-4.5 (6 members), SSP3-7.0 (5 members), and SSP5-8.5 (8 CESM2 and 1 RACMO2 members). One CESM2 SSP5-8.5 scenario includes ice dynamics (dyn). The black line represents the reanalysis-forced RACMO2.3p2 simulation (1958–2020)[10]. **b** Time series of anomalies in the upper-atmospheric global temperature (500 hPa) relative to pre-industrial (1850–1949). Thick coloured lines represent ensemble mean temperature anomalies for the same model scenarios described in **a**. The black line is based on ERA5 reanalysis[25], for which anomalies are estimated relative to 1950–1990. **c** same as **a** but for firn refreezing capacity. Coloured dots and whiskers estimate the timing and uncertainties of the GrIS-wide peak refreezing for SSP5-8.5 (red) and SSP3-7.0 (orange). **d** Correlation between SMB and the runoff line altitude; a cubic fit is used based on all simulations spanning the period 1850-2099, relevant statistics are listed. Surface mass loss (SMB < 0) is converted into sea level rise equivalent assuming that 362 Gt of ice raise global sea level by 1 mm. **e** Zoom out of **d** to highlight the intercept between the cubic fit and the threshold runoff line altitude (blue line and band).

6-hourly basis. Firn is initialised in January 1950 using snow temperature and density profiles from the offline Institute for Marine and Atmospheric research Utrecht-Firn Densification Model (IMAU-FDM)[46]. Ice albedo is prescribed from the 500 m Moderate-resolution Imaging Spectroradiometer (MODIS) 16-day surface albedo product (MCD43A3) as the lowest 5% annual values averaged for the period 2000–2015, and clipped between 0.30 for dark bare ice and 0.55 for clean bright ice under perennial firn. This model setting has been thoroughly evaluated and realistically represents the contemporary SMB of the GrIS, including the post-1990 mass loss acceleration[21]. Figures 1b–d, 2a–c, 3a–g and 5 show that the CESM2 forced historical data (1950–2014) are in close agreement with the state-of-the-art RACMO2.3p2 simulation (1958–2020)[10], forced by a combination of reanalyses including ERA-40 (1958–1979)[23], ERA-interim (1980–1989)[24], and ERA5 (1990–2020)[25].

## Statistical downscaling

Following Noël et al.[22], the CESM2-forced RACMO2.3p2 projections (1950–2099) and additional CESM2-based pre-industrial (1850–1949), historical simulations (1950–2014) and projections (2015–2300) are statistically downscaled from the model resolution of 11 km and ~111 km respectively to the ice mask and topography of the Greenland Ice Mapping Project (GIMP) Digital Elevation Model (DEM)[47] down-sampled to 1 km. In brief, the downscaling procedure corrects individual SMB components (except precipitation), i.e., primarily melt and runoff, for elevation and ice albedo biases on the coarse model grids. For the RACMO2 projections, runoff, melt and sublimation are corrected on the 1 km DEM using daily-specific vertical gradients, while CESM2 outputs used monthly values due to the unavailability of daily time series. Underestimation of melt and runoff due to locally observed ice albedo < 0.30 in the low-lying ablation zone, i.e., unresolved on the coarse model grids, is then minimised using a 1 km MODIS 16-day product averaged for 2000–2015. For instance, Supplementary Fig. 4 illustrates the downscaling procedure for year 2099 under SSP5-8.5. Supplementary Fig. 4a–c show the input topography, total precipitation and runoff on the native 111 km CESM2 grid, and Supplementary Fig. 4d–f the output fields after statistical downscaling to 1 km spatial resolution is performed.

For RACMO2, total precipitation, including snow and rainfall, is bilinearly interpolated from the 11 km onto the 1 km grid without additional corrections[17]. For CESM2 simulations, total precipitation

and snowfall are spatially adjusted using long-term reanalysis-based RACMO2 data at 5.5 km spatial resolution (1958–2020 average)[10] as,

$$Af_X = \frac{X_{5.5km} - X_{111km}}{X_{111km}} \quad (1)$$

$$PR_{adjusted} = PR_{111km} \times (1 + Af_{PR}) \quad (2)$$

$$SF_{adjusted} = SF_{111km} \times (1 + Af_{SF}) \quad (3)$$

where $Af_X$ is the spatial adjustment factor for variable X, i.e., total precipitation (PR; Supplementary Fig. 5a), or snowfall alone (SF; Supplementary Fig. 5b); $X_{5.5km}$ is the long-term (1958–2020) average of variable X simulated by reanalysis-forced RACMO2.3p2 at 5.5 km[10], and $X_{111km}$ is the historical (1950–2014) average of variable X simulated by CESM2 at 111 km. The adjustment factors were estimated for the present-day period showing negligible trends in total precipitation and snowfall ($\sim 0.1$ Gt yr$^{-2}$ for 1958–2020). Rainfall (RA) is then calculated as a residual following,

$$RA_{adjusted} = PR_{adjusted} - SF_{adjusted} \quad (4)$$

This spatial refinement is required since the smoothed topography in CESM2 at 111 km resolution (Supplementary Fig. 4a) underestimates (resp. overestimates) elevation at the GrIS margins (resp. in the interior) by up to 700 m relative to the 1 km GIMP DEM (Supplementary Fig. 5c). As a result, topographically-forced precipitation is underestimated and precipitation propagates too far inland, altering the spatial distribution and amount[26]. Note that the adjustment factors show similar patterns to the topography difference between the 111 km and 1 km grids (Supplementary Fig. 5). This first-order correction minimises the discrepancy between reanalysis-based and model-based precipitation. As a result, GrIS-integrated historical precipitation (704 ± 37 vs. 783 ± 41 Gt yr$^{-1}$) and snowfall (648 ± 26 vs. 720 ± 29 Gt yr$^{-1}$) are reduced by 10% in the adjusted product relative to the native CESM2 ensemble mean (12 members; 1950–2014). The adjusted GrIS-integrated precipitation is on par with previous benchmark reanalysis-forced RACMO2 simulations[21]. Refreezing (RF) is then estimated as a residual on the 1 km grid following,

$$RF = ME + RA - RU \quad (5)$$

where ME is the statistically downscaled surface melt, RA is the bilinearly interpolated (RACMO2) or adjusted (CESM2) rainfall, and RU is the statistically downscaled runoff at 1 km. Here we do not account for condensation at the firn surface, which is assumed to be negligible. Firn refreezing capacity (RFcap), i.e., the fraction of rain and meltwater effectively retained or refrozen in firn, is estimated as,

$$RFcap = \frac{RF}{ME + RA} \quad (6)$$

For the SSP5-8.5 scenario projection that accounts for ice dynamics beyond 2100 (i.e., glacier thinning and retreat), we follow the same approach as described above except that we let the ice sheet mask and surface topography fluctuate on annual basis in the fully coupled CESM2-CISM2. Statistical downscaling to 1 km uses monthly elevation gradients of SMB components estimated for a dynamic ice sheet. To account for glacier thinning and retreat on the 1 km grid, we combine the ice mask and surface topography from the GIMP DEM ($h_{surface}$), and bed topography from BedMachine v3[48] ($h_{bed}$) both down-

sampled to 1 km to estimate a reference ice thickness ($th^0$) as,

$$th^0 = h_{surface} - h_{bed} \quad (7)$$

We use ice thickness change modelled by CISM2 at 4 km resolution and down-sampled to 1 km ($\Delta h$) to estimate annual (y) ice thickness fluctuations as,

$$th^y = th^{y-1} + \Delta h \quad (8)$$

with $\Delta h = 0$ m for the initial year 2100. A dynamic surface topography at 1 km ($h_{dyn}$) is then estimated on annual basis as,

$$h_{dyn}^y = th^y + h_{bed} \quad (9)$$

If $th^y = 0$ m, $h_{dyn}^y$ is set to the bed topography and the corresponding grid cell is removed from the GIMP ice mask at 1 km, enabling glacier retreat. Precipitation components are adjusted on a monthly basis following Eqs. 1–4 on the dynamic ice mask. Supplementary Fig. 6 illustrates the statistical downscaling procedure that accounts for ice dynamics.

## Time series of SMB components
We find strong correlations ($R^2 = 0.88 - 0.97$) in GrIS-integrated SMB and components between the CESM2-forced RACMO2.3p2 projections (1950–2099) and the corresponding CESM2 historical (1950–2014) and SSP1-2.6/SSP5-8.5 forcing (2015–2099) (Supplementary Fig. 7). These high correlations also hold for the different sectors of the GrIS (Supplementary Fig. 8). Model differences between GrIS-integrated SMB and components are also small. Notably, CESM2 has lower melt (36 Gt or 5.2%) and runoff (31 Gt or 5.1%), combined with larger rainfall (33 Gt or 37%), resulting in larger refreezing (28 Gt or 4.5%). Note that the larger relative difference in rainfall originates from the mass flux being small compared to other components. Nevertheless, none of these model differences are statistically significant, i.e., differences are smaller than two standard deviations, highlighting the good agreement between the two downscaling methods. Uncertainty is estimated as one standard deviation of the spatially integrated SMB components for the corresponding period (Supplementary Table 2).

## Runoff line altitude
The runoff line altitude is estimated as the highest elevation with significant surface runoff, i.e., > 100 mm w.e. yr$^{-1}$ or an average of 1 mm w.e. per day in summer. This criterion was applied on an annual basis to all simulations statistically downscaled to the 1 km grid and for individual sectors of the GrIS. The runoff line altitude is estimated by averaging surface elevations within 5 mm w.e. runoff bins, and linearly interpolating elevation between the two runoff bins closest to the threshold of 100 mm w.e. yr$^{-1}$. We find high correlations (R = 0.79 – 0.97) in (regional) runoff line altitudes between the CESM2-forced RACMO2.3p2 projections (1950–2099) and the corresponding CESM2 historical (1950–2014) and SSP1-2.6/SSP5-8.5 forcing (2015–2099) (Supplementary Fig. 9). The correlation between CESM2 and RACMO2.3p2 can differ in the historical and scenario periods (Supplementary Fig. 9). This results from the coarse resolution of CESM2 that propagates runoff too far inland the ice sheet, leading to an overestimated runoff line altitude in the historical period. This resolution artefact becomes less pronounced as runoff is produced further inland following atmospheric warming in the scenario projections. To address this, we estimate linear regression coefficients, i.e., regression slope (a) and intercepts (b), to adjust the runoff line altitude in each sector and GrIS-wide. Regression coefficients based on the historical data (1950–2014, grey dots in Supplementary Fig. 9) are used to correct the 11 CESM2 pre-industrial (IND) and 12 CESM2 historical simulations (HIST), whereas regression coefficients from the full data set

(1950–2099, grey and black dots in Supplementary Fig. 9) are used to correct the 25 CESM2 scenario projections (SSP). Uncertainty in the runoff line altitude is estimated as two standard deviations for a given period. Concerning the threshold altitude, uncertainty is first estimated for SW Greenland as 2 standard deviations around the average runoff line altitude in 2080–2099 (Supplementary Fig. 2a), i.e., marking the regional peak refreezing with 90 ± 10% of the sector area experiencing significant runoff (>100 mm w.e. yr$^{-1}$). For other sectors and the GrIS, we combine surface hypsometries with the 90 ± 10% area threshold to estimate a lower (80%) and upper (100%) uncertainty bound (blue line and band in Supplementary Fig. 2b–h and Supplementary Table 1).

## Timing of peak refreezing

To time the crossing of (regional) peak refreezing under SSP5-8.5 and SSP3-7.0, we fit a quadratic curve to the ensemble-mean runoff line altitude for 1950–2099. A quadratic regression is justified as the (regional) runoff line altitude fluctuates non-linearly in time for both projections (Supplementary Fig. 3). We find that in all sectors, quadratic curves estimated for SSP5-8.5 scenarios show excellent agreement with our long-term SSP5-8.5 scenarios extending beyond 2100. We hypothesise that this approach is also valid under SSP3-7.0, for which a similar non-linear evolution of the runoff line is obtained prior to 2100. The intercept between the quadratic curves and the threshold runoff line altitude with uncertainty bands, i.e., for which 90 ± 10% of the GrIS or individual sector area produces significant runoff (>100 mm w.e. yr$^{-1}$), estimates the timing of the peak refreezing and associated lower and upper uncertainty bounds (Supplementary Fig. 2 and Supplementary Table 1). Note that this timing uncertainty results from the approach described above and does not account for internal model uncertainty.

## Data availability

The statistically downscaled SMB data sets at 1 km (1850–2300) generated in this study have been deposited on Zenodo [https://doi.org/10.5281/zenodo.7100706].

## Code availability

The statistical downscaling technique is presented in Noël et al.[22]. The regional climate model RACMO2.3p2 is presented in Noël et al.[10,38]. The earth system model CESM2 is presented in van Kampenhout et al.[26]. The CESM2-forced RACMO2.3p2 data are presented in Noël et al.[21,27].

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

## Acknowledgements

B.N. was funded by the NWO VENI grant VI.Veni.192.019. M.R.v.d.B. acknowledges support from NWO/ALW, NESSC and PROTECT. This project has received funding from the European Union's Horizon 2020 research and innovation programme under grant agreement No 869304, PROTECT contribution number 45. The CESM project is supported primarily by the National Science Foundation (NSF). This material is based upon work supported by the National Center for Atmospheric Research (NCAR), which is a major facility sponsored by the NSF under Cooperative Agreement No. 1852977. Computing and data storage resources, including the Cheyenne supercomputer (https://doi.org/10.5065/D6RX99HX), were provided by the Computational and Information Systems Laboratory (CISL) at NCAR.

## Author contributions

B.N. designed the study, prepared the manuscript, conducted the RACMO2 projections and statistically downscaled the presented data sets. B.N. and M. R. van den Broeke analysed the data. J.T.M.L., W.H.L. and K.T.-C. provided the CESM2 data sets. All authors commented on the manuscript.

## Competing interests

The authors declare no competing interests.
