## [Peer Review File · Nature Communications]

Peak refreezing in the Greenland firn layer under future warming scenariosReviewers' comments:

Reviewer #1 (Remarks to the Author):

KEY RESULTS

This manuscript presents analysis of 20th, 21st, and 22nd century evolution of surface mass balance across the Greenland Ice Sheet, focusing on the infiltration and refreezing of surface meltwater within firn. Key results include snow, rain, meltwater runoff, and surface meltwater infiltration and refreeze across the GrIS for the 20th (1950-2014) and 21st (2015-2099) centuries. Model results were generated using the Community Earth System Model (CESM2) run for various historical simulations and Shared Socioeconomic Pathway emission scenarios, and dynamically downscaled using the Regional Atmospheric climate model version 2.3p2 (RACMO2.3p2). The modeled evolution of GrIS surface mass balance during the 21st century shows a nonlinear response to climate warming as the ablation zone expands and the firn zone retreats. Authors analyze model output to define the runoff line, the uppermost altitude where surface meltwater is at least 100 mm w.e. yr⁻¹. This runoff line advances upward in elevation nonlinearly as GrIS meltwater runoff increases due to snowfall rates remaining relatively stable (which precludes firn replenishment), while rain and surface meltwater increase, the firn column saturates with meltwater refreeze, and the areal extent of the firn zone shrinks. The authors fit a quadratic to the runoff line altitude increase per time from 1950-2099 according to various climate scenarios. These quadratic fits, applied to GrIS hypsometry, are used to project the timing when more than 90% of the GrIS is below this runoff line altitude. Results for high emission scenarios show that this so-called collapse of firn refreezing capacity happens sooner in the 22nd century (2148) than low emission scenarios.

VALIDITY

The topic of meltwater refreeze within Greenland Ice Sheet firn is of extreme importance to glaciologists, but I am not sure that the work presented here is sufficiently insightful to warrant the kind of public and media attention garnered by Nature Climate Change. What do glaciologists gain from reading this paper? If the current rate of firn saturation due to refreeze keeps up, then the firn will eventually saturate. That saturation point happens in the 22nd century, sooner (2148) that century for high emission/high melt scenario than low emission/low melt scenarios. The quality of this result is consistent with what experts expect. The specific timing might be useful but the method for determining this timing requires more explicit explanation and physical justification of why it is valid. Why is fitting a curve to firn zone shrinkage 1950-2099 accurate for extrapolating that process further inland where the firn column is presumably colder and thicker? Why is this curve, which was forced by 1950-2099 climate, applicable to describe Earth system conditions beyond 2100?

ORIGINALITY AND SIGNIFICANCE

Originality is the good performance/agreement between model output and observations 1950-2014. But this was already reported in (Noël and others, 2019). Projecting the timing and extent of firn saturation due to meltwater infiltration and refreeze, as well as the insight into mass balance partitioning (snow, rain, runoff) for 1950-2099 under Shared Socioeconomic Pathway emission scenarios is significant, I think, because of the downscaling to 1 km. Other applications of the CESM2 for GrIS projections (Muntjewerf and others, 2020) apparently do not achieve this high resolution nor employ recently improved representation of the Greenland firn layer (Ligtenberg and others, 2018).

UNCERTAINTY TREATMENT

Uncertainty is never very clearly defined in this paper, but the reader can deduce that the presented uncertainty (line 113) is defined as the range of modelled outcomes. This range arises from the different climate scenarios. It therefore reflects spread in the modeled climate forcing, but not necessarily in ice sheet surface mass balance response. Uncertainty due to missing physical processes in the model is not clearly reported. "Uncertainty" appears first on line 113 without clear definition. Uncertainty shown on Figure 3, 4, and 5 is similarly derived from climate model scenario comparisons. The supplement does not expand on uncertainty treatment.

The authors acknowledge a few processes that are not accounted in this modeling exercise (lines

147-148) yet I was left wondering about what magnitude would unresolved processes (e.g. firn aquifers, firn aquifer drainage, elevation-lowering-feedback) alter the reported results. Perhaps previous work could be leveraged to estimate: How much of the modelled firn column could conceivably be occupied by aquifer (Forster and others, 2014)? How much water in the aquifer could drain (Poinar and others, 2017)? How much extra surface melt could be generated were the GrIS model adjusted to show ice sheet thinning and subsequent surface elevation lowering (Muntjewerf and others, 2020)?

CONCLUSIONS

I find the conclusions questionable. For example, Line 166 "Greenland surface mass loss increases non-linearly with the advance of the runoff line, irrespective of the warming scenario" This conclusion, as applied to projections past 2099, is based on a nonlinear quadratic fit to 1950-2099 output then used to project the advance of the runoff line. Therefore, does it necessarily reflect real world future evolution of the GrIS firn meltwater storage capacity? Rather, the conclusion could simply be an artifact of the methodology and the assumption baked into that approach, namely that the saturation and runoff line advance will continue to march upward 2099 onward at the same nonlinear rate modeled for 1950-2099. Why is that assumption valid?

LIMITATIONS

I do not work with climate or surface mass balance models on a regular basis and therefore might be missing some common knowledge shared within that community of practice.

To make the work more readily accessible to a wider readership, I suggest that the following technical details regarding the model be presented outright:

- vertical height of the firn column represented in the model, i.e. 40m depth? 100m+ depth?
- spatial variability in the vertical height of the firn column represented in the model, i.e. how does this vary across the ice sheet and does model show the firn thinning as well as losing areal extent?
- agreement or disagreement between the chosen regional climate model (RACMO2.3p2) with other regional climate models (HIRHAM5, MARv3.9)
- explain how the statistically downscaled refreezing (line 254), which is calculated as a residual derived from surface melt, rainfall, and runoff, compares to refreezing calculated within firn/snow models thermodynamically constrained (i.e. output prior to downscaling)
- similarly, how does the downscaled refreeze residual (equation 6) compare to independent, direct model output of refreeze? Some validation, and perhaps comparison with other firn models (Stevens and others, 2020), might elucidate the robustness of this Refreezing (RF) estimate.

LINE COMMENTS

Title. Consider editing the title to include the word "Projected" to reflect methodology and honor uncertainty associated with forecasts: "Projected 22nd century tipping point in refreezing accelerates Greenland ice sheet mass loss"

Line 2. Instead of "collapse of firn refreezing" consider referring to it as "firn saturation" or something that evokes the physical process more directly. Reaching the saturation capacity of the firn is less something collapsing/tumbling/falling/degrading and more a process of saturation/freezing/solidifying/squeezing out pore space.

Line 3. Consider stating which Earth System Model outright in the abstract.

Line 6. Consider "project" instead of "predict"

Line 6. Consider "could reach saturation" instead of "could collapse"

Line 11. Consider "likely avoids firn saturation altogether" instead of "likely prevents"

Line 16-17. Consider editing to "These retention mechanisms prevent meltwater runoff, mitigating surface mass loss."

Line 28. Consider "In a warming climate, exceeding firn saturation and refreezing capacity is inevitable" instead of "collapse of firn refreezing"

Line 64. Consider adding some brief explanation of how RACMO2.3p2 compares to other regional

climate models and why it was chosen for this work.

Line 65. Consider changing "we predict" to "model output show the firn zone retreating xx% from current (2020) extent to cover 61%"

Line 68. As it is written, the non-linear SMB decline could be explained by the expansion of ablation zone and bare ice where there is zero firn meltwater storage. Consider editing to fit firn zone into this explanation.

Line 73. Consider specifying "the remainder of the precipitation input is discharged"

Line 84. Consider justifying why this quadratic fit to model output is physically reasonable for the extrapolation.

Line 87. Consider changing the title to something that speaks to physical process, i.e. "Firn saturation coupled with insufficient snowfall yield tipping points"

Line 91. Consider editing to "In the percolation zone, which is the zone extending just above the equilibrium line altitude where some surface melt occurs, refreezing significantly declines" This definition agrees with Cuffey and Paterson (2010) and is more descriptive than "low accumulation zone"

Line 100. Consider highlighting this insight in the abstract. Results show that expansion of the firn saturation zone is due not only to meltwater refreeze but also lack of snow to replenish firn.

Line 127. Consider editing to "Equation 1 can be extrapolated to predict because X" Questions: How does equation 1 adequately capture 22nd century nonlinearities when it was not trained on 21st century data? Would another inflection point be expected that would call for another, new, different set of coefficients? Alternatively, would it take longer (slower RLA hike) in 22nd century for thicker, colder firn column to saturate progressively inland?

Line 162. Consider adding a sentence to elaborate on how this study compares to citation 23, a study where the GrIS is dynamically coupled to CESM (i.e. surface elevation adjusted)

Line 166. "advance of the runoff line" evokes march toward the terminus. Consider instead "hike" or "increase of the runoff line"

Line 198. How deep does the 40-layer snow module extend? How does this compare to the observed firn column depth?

Line 292. Correct date on citation Poinar et al. (2017)

REVIEW REFERENCES

Cuffey, K.M., and Paterson, W.S.B., 2010, *The Physics of Glaciers*: Elsevier, Oxford.

Forster, R.R., Box, J.E., van den Broeke, M.R., Miège, C., Burgess, E.W., van Angelen, J.H., Lenaerts, J.T.M., Koenig, L.S., Paden, J., Lewis, C., Gogineni, S.P., Leuschen, C., and McConnell, J.R., 2014, Extensive liquid meltwater storage in firn within the Greenland ice sheet: *Nature Geoscience*, v. 7, no. 2, p. 1–4.

Ligtenberg, S.R.M., Munneke, P.K., Noël, B.P.Y., and Van Den Broeke, M.R., 2018, Brief communication: Improved simulation of the present-day Greenland firn layer (1960-2016): *Cryosphere*, v. 12, no. 5, p. 1643–1649.

Muntjewerf, L., Petrini, M., Vizcaino, M., Ernani da Silva, C., Sellevold, R., Scherrenberg, M.D.W., Thayer-Calder, K., Bradley, S.L., Lenaerts, J.T.M., Lipscomb, W.H., and Lofverstrom, M., 2020, Greenland Ice Sheet Contribution to 21st Century Sea Level Rise as Simulated by the Coupled CESM2.1-CISM2.1: *Geophysical Research Letters*, v. 47, no. 9.

Noël, B., van Kampenhout, L., van de Berg, W.J., Lenaerts, J., Wouters, B., and van den Broeke, M., 2019, Brief communication: CESM2 climate forcing (1950–2014) yields realistic Greenland ice sheet surface mass balance: *The Cryosphere Discussions*, v. 2, p. 1–17.

Poinar, K., Joughin, I., Lilien, D., Brucker, L., Kehrl, L., and Nowicki, S., 2017, Drainage of southeast Greenland firn aquifer water through crevasses to the bed: *Frontiers in Earth Science*, v. 5, no. February, p. 1–15.

Stevens, C.M., Verjans, V., Lundin, J., Kahle, E., Horlings, A., Horlings, B., and Waddington, E., 2020, The Community Firn Model (CFM) v1.0: *Geoscientific Model Development Discussions*, no. January, p. 1–37.

Reviewer #2 (Remarks to the Author):

In '22nd century tipping point in refreezing accelerates Greenland ice sheet mass loss', Noël et al. seek to investigate the limits of the buffering effect from meltwater refreezing in the Greenland Ice Sheet's percolation zone on the overall ice sheet surface mass balance. The authors achieve this by interrogating and downscaling a variety of climate model products, both directly from a CESM (Community Earth System Model), and from a regional climate model (RACMO) simulation, forced by CESM output through the 21st century. Through this process, the authors find that under all emissions scenarios, refreezing on the Greenland Ice Sheet (GrIS) increases through the 21st century as surface melt increases in magnitude in the percolation zone and also occurs at higher elevations. Under the high emissions (SSP5-8.5) scenario, model outputs indicate that the runoff limit increasingly extends inland through the 21st century. Using this as a reasonable proxy for refreezing, extrapolation of the trend implies that refreezing is likely to decline some time in the mid-22nd century. This marks a tipping point for this region of the ice sheet, whereby surface melt is partitioned more and more in to runoff (cf. refreezing), and contributes to sea level rise.

Refreezing in the GrIS' percolation zone is central to understanding the fate of meltwater, and with it the contributions to sea level rise. The numbers demonstrate the importance: nearly one half of ice sheet melt refreezes in place. The paper topic is therefore highly relevant to a critical issue in projecting ice sheet mass balance. The concept of a 'tipping point' is not particularly new, as the authors have illustrated the process in an earlier paper focused on Greenland's glaciers and ice caps (Noël and others, 2017). But the GrIS tipping point has not yet been investigated. I appreciate the effort by the authors to leverage a number of climate products; their results are not particularly skewed to a single simulation. The importance of the subject and use of a suite of model results are all reasons that the manuscript warrants publication. However, I remain quite skeptical of the results, which I believe are highly speculative for a few reasons:

- 1) The manuscript's core take-homes rely on extrapolation of model results out some 50-300 years.
- 2) Critical uncertainties in the buffering capacity of the percolation zone persist, but these are largely related to shortcomings in our understanding of the infiltration process. How impermeable are 'ice slabs'? How deep does meltwater infiltrate, and how do we parameterize complex fingering flow in models that cannot treat such local & heterogeneous processes? Not only are these uncertainties not addressed or overcome in this paper, but the opposite is true -- the firn model in the CESM runs is even more simplistic than the tipping bucket model implemented in RACMO.
- 3) The CESM output is nominally on a 111 km grid. On this scale, the percolation zone in many regions around the ice sheet is ~ 1 grid cell. The authors overcome this limitation with statistical downscaling, but the downscaling is performed using historical or present-day data and are assumed to remain static in time. For instance, despite the large increase in mass loss reported, all modeling assumes that the ice sheet does not change in elevation over the next century.

Considered in aggregate, it's unclear how these uncertainties influence the results. But I suspect the true timescale for the ice sheet to reach a tipping point has an uncertainty envelope that is much larger than the ± 15 years the authors report based on the model suite.

The primary contribution of the manuscript is in projecting a timescale of this percolation zone 'tipping point'. Since this is the case, I believe the fidelity of the model output is of the utmost importance. Because I remain skeptical of the model output, especially in the absence of any discussion of the sources of uncertainty and the plausible impact on the interpretation, I cannot recommend the manuscript for publication.

I recognize that the model outputs are what they are, and to some degree cannot be improved without advances in understanding of the processes controlling meltwater infiltration in firn. But I believe the paper could still be improved if the authors removed the extrapolation and more appropriately acknowledged and discussed the sources of uncertainty in the model outputs. Of course in doing so, the primary take-homes may be far more uncertain, not quite as confidently impactful, and therefore perhaps less suitable to a high-profile journal.

In light of my recommendation, I have not provided detailed comments on the manuscript. But there were three other details of the paper that I'll comment on, and which I believe would improve a revised paper if fixed.

I found that the introduction did not motivate a specific problem which the paper set out to solve or advance. The authors acknowledge 'major uncertainties' (line 26-27) in future meltwater refreezing, but fail to communicate what these uncertainties are. If this study overcomes any of these uncertainties, then outline them in the introduction to give the reader a reason to invest time in the paper. Incidentally, the authors state in the next sentence (line 28) that collapse of firm refreezing is inevitable. If this is the case, the authors are stating that the reader shouldn't be investing time in the paper to begin with because we should already know the main conclusion!

To me, the word 'collapse' implies a rapid time scale and complete disintegration. But I believe the results show neither of these. The results indicate a tipping point in refreezing in the SW sector by the end of the 21st century, but certainly not a rapid extinction of refreezing. The primary model take-home is based on extrapolation of a proxy for this tipping point. Don't get me wrong, I believe the ice sheet is in a major state of change, and this shouldn't be minimized by cautious scientists. But I interpreted the statements of 'collapse' to be an exaggeration that raised red flags with the results.

The authors define capacity as the amount of water that has been retained or refrozen (eqn 7), but I believe a more accurate description of this quantity is 'refreezing fraction'. Capacity is the amount of pore space that is available for meltwater refreezing/retention (e.g. see methods in Harper et al., (2012)) and does not depend directly on melt amounts. When the authors describe/present a reduction in refreezing capacity (e.g. Figure 1), this should be termed refreezing fraction. The capacity would be the air pore space integrated over the ice sheet.

References

- Harper J, Humphrey N, Pfeffer WT, Brown J and Fettweis X (2012) Greenland ice-sheet contribution to sea-level rise buffered by meltwater storage in firn. *Nature* 491(7423), 240–243 (doi:10.1038/nature11566)
- Noël B, Van De Berg WJ, Lhermitte S, Wouters B, Machguth H, Howat I, Citterio M, Moholdt G, Lenaerts JTM and Van Den Broeke MR (2017) A tipping point in refreezing accelerates mass loss of Greenland's glaciers and ice caps. *Nat. Commun.* 8(9296) (doi:10.1038/ncomms14730)

Reviewer #3 (Remarks to the Author):

Dear Dr. Noël et al.,
Thank you for an interesting study.
See my comments in the attached file.
Best regards,
Baptiste Vandecrux

***Review of "22nd century tipping point in refreezing accelerates Greenland ice sheet mass loss" by Noël et al.
Baptiste Vandecrux***

Review conducted using the Nature guidelines: <https://www.nature.com/ncomms/journal-policies/guide-to-referees>

General comments:

What are the major claims of the paper?

The manuscript presents simulations from an Earth System Model, CESM2, and a Regional Climate Model, RACMO2.3p2, over Greenland during 1950-2100 and under various Shared Socioeconomic Pathways (SSP) scenarios. The study describes on the evolution of refreezing, a key component of the Greenland ice sheet mass balance. A “tipping point” in meltwater refreezing is simulated in ~2080 in the Southwest (SW) region in scenarios SSP5-8.5 and is concurrent to a runoff area reaching ~90% of that region in 2080. Runoff line altitude is thereafter used as a proxy to describe when runoff area reaches 90% of a given region and when meltwater refreezing is expected to reach this tipping point. Eventually, an extrapolation method is used to predict when this tipping point is reached in different regions and for the ice sheet as a whole beyond 2100, when no simulation is available.

Are the claims novel? If not, please identify the major papers that compromise novelty

It is the first time, to my knowledge, that the SMB components from SSP-forced CESM2 simulations are being downscaled to 1 km and adjusted for biases. This makes the presented model outputs very robust over the 1950-2100 period with both high spatial resolution and good match with available observations. The tipping point in refreezing in SW during the 21st century is also novel to my knowledge and relies on robust simulations. The extrapolation of future runoff line altitude, runoff area and crossing of a tipping point in refreezing are also new.

Will the paper be of interest to others in the field? Will the paper influence thinking in the field?

Yes

Are the claims convincing? If not, what further evidence is needed?

Despite the value of the presented model runs (product for all SSP, high spatial resolution, adjusted to observations during the historical period...), I have some limitations about the main result of the study: the prediction of a year in the 22nd century when meltwater refreezing reaches a tipping point and collapses. There are two problems with this claim: its wording and its robustness.

For the wording, I have an issue with the use of two words: “tipping point” and “collapse”.

- Dictionaries define “tipping point” as “the point at which an object is no longer balanced and adding a small amount of weight can cause it to topple”. It carries both the idea of imbalance and irreversibility. Here, the phenomenon that is described as a tipping point is the refreezing total reaching a maximum and then starting to decline (Figure 2b). Nothing is said about how irreversible that curve inflexion is. If late efforts are made for climate mitigation, would the snow and firn layer be able to recover and refreezing start to increase again? This is not mentioned in the manuscript and therefore I find the term “tipping point” used without the appropriate justification.
- Dictionaries define "Collapse" by "to fall down suddenly or completely". Here the "tipping point" in refreezing that estimated either for the GrIS or for its subregions characterizes the moment when meltwater refreezing start to

decrease. This is defined from what is simulated in the SW region in ~2080. But looking at Figure 2b, it is very subjective how to describe the slope inversion that occurs for the SSP5-8.5 scenario, in 2080 and in the SW region. There is indeed a decline of refreezing, but this decline is neither complete (there are still large amounts of meltwater being refrozen in 2099) nor more abrupt than over other periods of the 21st century (see temporary decreases of refreezing in 2045 and 2060 in the CESM2-RACMO run, Figure 2b). The choice of word “collapse” should therefore be better justified.

The second major issue is relative to the very optimistic uncertainty bounds provided with the year of expected crossing of the tipping point. At the moment, the uncertainty bounds are calculated (as I understand it) from the spread with which SSP scenarios within each ensemble cross the critical runoff line altitude. This uncertainty estimation does not account for 1) the limitations of the assumptions being used and 2) the uncertainty of the extrapolation method.

- Regarding the solidity of the assumptions being used. First the tipping point is defined as the year when refreezing initiates a long-lasting decline (Figure 2b), then this definition is translated into the year when 90% of a region experiences runoff (l. 120-122), eventually it is translated into the year when the runoff line altitude reaches a region-specific critical altitude (Figure 4). The study currently fails to provide an estimation of uncertainty at each conceptual leap. For another region than SW, what is the chance to have its refreezing decreasing before the 90% of its area suffers runoff? How is the uncertainty on this 90%, +/-10%, is estimated and how is it used in within the uncertainty estimation of the year of tipping point? What is the chance for the hypsometric relationship used to relate runoff area and runoff altitude, to be the same in the 22nd century as the one used in Figure 4?
- Regarding the extrapolation method, what is the confidence envelope that applies to the quadratic functions presented? This is usually provided by curve-fitting algorithm as uncertainty bounds on the polynomial's coefficients. Even more uncertain: why would the temporal evolution of runoff line altitude be quadratic? What would happen if a linear function, exponential function or polynomials of higher degree were used instead? The study currently does not address this uncertainty on the future trajectory of the runoff line altitude.

Considering these two sources of uncertainty, I find it very optimistic to present the year when the “Greenland firn refreezing capacity could collapse” with an uncertainty of “±15” years.

Are there other experiments that would strengthen the paper further? How much would they improve it, and how difficult are they likely to be?

Better description and quantification of the uncertainty sources and investigation of multiple extrapolation functions could strengthen the manuscript. The authors should describe better how the decrease of meltwater refreezing they see in the South-West region qualifies as tipping point or collapse.

Are the claims appropriately discussed in the context of previous literature?

Something I am curious about is whether this maximum refreezing in SW Greenland was also seen in previous versions or RACMO such as van Angelen et al. (2013). van Angelen et al. (2013) became the reference model study for firn pore space loss and threatened firn refreezing capacity in Greenland. It could be beneficial to the study to specify how these better models and climate scenarios improved the predictions for meltwater refreezing in Greenland for the 21st century.

Additionally, a similar tipping point was described by the authors in Canada and Greenland's glaciers and ice caps (Noël et al., 2017, 2018). Maybe they could be discussed a bit more to highlight the similarities and differences with the Greenland ice sheet. They could also be used to build trust in the chain of assumptions linking the crossing of a tipping point in refreezing to the runoff line reaching a certain critical altitude. Indeed, the single SW region is currently used as basis for describing the tipping point in meltwater refreezing and the SW case is extrapolated in other regions and for the entire ice sheet. Do Greenlandic and Canadian glaciers and ice caps show a similar relationship between tipping point in refreezing, area experiencing runoff and runoff line reaching a critical altitude? These Greenlandic and Canadian glaciers and ice caps can be used to assess the robustness of the approach and better quantify its uncertainty. This is just a suggestion and is not necessary if a robust uncertainty assessment can be presented from the SW region only.

Is the manuscript clearly written? Yes

Could the manuscript be shortened to aid communication of the most important findings? No, it is already straight to the point.

Have the authors done themselves justice without overselling their claims?

I think a bit more caution with regards to the wording and to the uncertainty estimation would definitely make the work more robust.

Have they been fair in their treatment of previous literature? See the two minor points above.

Have they provided sufficient methodological detail that the experiments could be reproduced? Yes

Is the statistical analysis of the data sound? The analysis is sound apart from the uncertainty estimation in the extrapolation.

Should the authors be asked to provide further data or methodological information to help others replicate their work? Yes

As a concluding remark, I would like to highlight the quality and the value of the model outputs presented over the 1950-2099 period. The extrapolation into the 22nd century is an interesting preview of what may happen in the future but is rather uncertain to be the main result highlighted by the study (and its title). Potentially, refocusing the study on the tipping point simulated in SW Greenland, and keep the extrapolation as an interesting discussion point, could be a way to make the study more robust.

l. 1: "90%", in line 52 it is 92%

l. 2: "the firm covering..." Is the ELA considered equivalent to the firm line? This is neglecting the superimposed ice area. It should be mentioned. Alternatively, the sentence could be rephrased to state that 92% of the ice sheet is accumulation area (Fig.1a) and that a majority of that is firm.

l.67-71: Consider cutting that sentence in two.

l. 74-79 (related to l.257-259): The refreezing capacity is usually defined as the maximum amount of water that can be refrozen within the firm (i.e. the refreezing that the firm is “capable” of). Here it is defined as what is actually refrozen divided by the sum of two factors, melt and rainfall, which are two external factors to the firm. Maybe it should be called refreezing ratio, fraction or index?

l. 77: The point above becomes problematic: Since your refreezing capacity is currently calculated from refreezing which is calculated (in the downscaled product) from runoff (Eq. 6 and 7), it is the increased runoff that causes the decrease of your calculated refreezing capacity, not the other way around. If the refreezing capacity was defined from pore space and cold content, then the sentence would be accurate, but that would not fit with equation (5) anymore.

l.85-86: Specify that the time in Eq. 1 is in years. Consider changing “x” to “t” or “yr” (optional). Considering the importance of equation 1 (along with the coefficients in Table S2) in the determination of the tipping point, I would expect to see a better illustration of these extrapolation functions in the main text. A possibility would be to bring Figure 5a earlier, make the historical runoff line altitude and fitted function more visible (no need to display after 2200?) and bring Table S2 in the main text. The new figure could also display as shaded area the ensemble spread (current uncertainty estimation) but also other types of uncertainties such as the confidence envelope of the quadratic fit or other fitting functions that could equally fit the historical data.

l. 101: “predicted to cross...” Although it was mentioned in the sentence just before, please mention in the same sentence that this result is linked to one or all scenario (e.g. “for these scenarios” or “for SSP5-8.5”).

l. 109: Same as above, please mention that this is only for the SSP5-8.5 scenarios. Please check the manuscript throughout so that scenario-specific results are always accompanied with a mention of the scenario they relate to.

l. 116: How is the uncertainty margin of 10% determined?

l. 117: “1984 +/- 118 m a.s.l., Fig. 3b” please add that this is the runoff line altitude for the period 2080-2099 (?)

l. 129-133: I believe that these results should come with a brief reminder of the assumptions they rely on. Something like “under the conditions that the temporal dynamic of runoff lines remains the same into the 22nd century”. These assumptions should also be discussed in more details elsewhere.

l. 152: Please mention the uncertainty on the climate evolution in the 22nd century, the uncertainty applying to the coefficients of the quadratic function used for extrapolation and the uncertainty applying to the shape of that extrapolation function.

l.167-171: This extrapolation is also daring. Using a quadratic function fitted to SMB values in the range [-800, 400] Gt yr⁻¹ and extrapolating it to 4655 Gt yr⁻¹ is rather uncertain. What is the confidence envelope that come with this fit? What would other functions (piecewise linear, exponential, polynomials of other degree) give? I am not sure the manuscript gains much from this analysis.

l. 175: “a CMIP6 model” maybe rephrase “a model member of CMIP6” or alike.

l. 189: “despite” makes the “higher climate sensitivity” sound like a bad behavior. Is it? Could you detail a bit more (if it is relevant)?

l. 188: I only screened reference 23 and thought that it presented only SMB and not its components. If the SMB components were presented before can you make clear how the outputs presented here are different from the ones presented in Muntjewerf et al. (2020). Note that this entry should be updated in the reference section.

l.192: “one SSP5-8.5 (2015-2099) CESM2 member”. Please specify which member for reproducibility. It should be also explained whether all members within SSP5-8.5 are equivalent (just alternative scenarios) or if within that ensemble they range from lower warming (SSP5?) to stronger warming (SSP8.5?). In the second case, it is even more important to state which member was used to force RACMO2.

l.220: I am a bit puzzled by the "the downscaling procedure corrects ... for ... biases". Is the downscaling procedure aiming to reduce the biases of a certain model output with regard to a certain dataset considered more reliable (that is what is done later with precipitation)? or is it just an adjustment method that keeps constant the amount of melt seen in a coarse cell, but within that cell allocates more melt to lower elevation and less melt in higher elevation? In the second case, I am not sure we can talk about "correction" or "bias" since no reference dataset is being used. In the first case, the dataset that is used as reference for the correction should be specified.

l. 220-221 “individual SMB components (except precipitation), i.e. primarily melt and runoff”, in the next line you add sublimation, maybe replace by a simpler listing of the downscaled variables.

l. 231: So the precipitation and snowfall in CESM (but not in CESM2-forced RACMO) are being adjusted to match the reanalysis-forced RACMO output. As I understand, the melt and runoff are being downscaled using topography and MODIS albedo. Can you explain why you don't use topography to adjust the precipitation in CESM or why you don't adjust melt and runoff in CESM2 to match reanalysis-forced RACMO just like is done for precipitation?

l.252-253: “reanalysis-forced RACMO2” In Figure S4b, the caption states that it is the CESM2-forced RACMO2.3p2 output (not reanalysis-forced). Which one is used?

l.257 and 279: Do I understand right that the CESM2 output is adjusted to the reanalysis-forced RACMO2 output first in the downscaling step and then a second time when the SMB components are integrated over specific regions? Is the second adjustment necessary?

It is legitimate to use the historical part of CESM2-forced RACMO2 output to adjust the CESM2 SMB components and runoff line altitude. But it should be clear throughout the manuscript when it is the CESM2 output that is being used or a series that was adjusted to CESM2-forced RACMO2. It also adds to my discussion point on uncertainties: how would the extrapolation functions look on the non-adjusted runoff line altitude time series?

Figure 1cd: It is hard for me to see the green line.

Figure 2: In the caption, at the end of the 7th line it should be “d Refreezing...”. In the following line: “CESM2-forced RACMO2.3p2 projection”, please add which SSP scenario is used.

Figure 3:

- Color of the minimal firn area: Can you use a color that is not already part of the runoff colormap?
- The use of “tipping point” in the legend: It seems here that several quantities go into "tipping point": A year when refreezing start to decline (Fig 2), an area that experiences runoff (Figure 4) or a "critical runoff line altitude" (Fig.3).

I think the legend here should show "critical runoff line" and the caption should tell briefly why that line is "critical" and how it relates to the tipping point in refreezing.

- "In a-f, the blue contour represents the critical runoff line altitude and associated uncertainty (dashed blue) marking the GrIS-wide tipping point in firn refreezing." I see the dashed blue only in (a), is that normal? I also find it hard to distinguish these dashed lines from the elevation contour.
- "The cyan area outlines the remaining firn area after the tipping point is passed." Please be more specific: In which scenario? In which year? If there are still some firn remaining, how does it qualify as a collapse or tipping point?

Figure 5: How is calculated the uncertainty that applies to the year when CESM2-forced RACMO simulation reaches the critical runoff line altitude (black dot with whiskers)? I have understood that the uncertainty in the crossing of this threshold was derived from the spread seen within each ensemble. For the CESM-forced RACMO2 simulation, there is only one member being used (1.192).

References:

- Muntjewerf, L., Petrini, M., Vizcaino, M., Ernani da Silva, C., Sellevold, R., Scherrenberg, M.D., Thayer-Calder, K., Bradley, S.L., Lenaerts, J.T., Lipscomb, W.H. and Lofverstrom, M., 2020. Greenland Ice Sheet Contribution to 21st Century Sea Level Rise as Simulated by the Coupled CESM2. 1-CISM2. 1. *Geophysical Research Letters*, 47(9), p.e2019GL086836.
- Noël, B., van de Berg, W.J., Lhermitte, S., Wouters, B., Schaffer, N. and van den Broeke, M.R., 2018. Six decades of glacial mass loss in the Canadian Arctic Archipelago. *Journal of Geophysical Research: Earth Surface*, 123(6), pp.1430-1449.
- Noël, B., van De Berg, W.J., Lhermitte, S., Wouters, B., Machguth, H., Howat, I., Citterio, M., Moholdt, G., Lenaerts, J.T.M. and van den Broeke, M.R., 2017. A tipping point in refreezing accelerates mass loss of Greenland's glaciers and ice caps. *Nature Communications*, 8(1), pp.1-8.
- Van Angelen, J.H., M. Lenaerts, J.T., Van den Broeke, M.R., Fettweis, X. and Van Meijgaard, E., 2013. Rapid loss of firn pore space accelerates 21st century Greenland mass loss. *Geophysical Research Letters*, 40(10), pp.2109-2113.

Dear editor and reviewers, please find our point-by-point responses to the reviewers' comments in blue. Since the new manuscript has been substantially modified compared to the previously reviewed version, some concerns raised by the reviewers are no longer relevant. Therefore, we listed below the major changes applied to the manuscript. We think that our additional data sets and analyses have greatly improved both the robustness and the impact of the results.

Major changes compared to NCOMMS-20-33873-T

Here you will find a list of major changes (A-E) that were applied to the new manuscript. These changes were inspired by the original reviewers' comments. Many of their concerns/comments are addressed/resolved in the new manuscript and are therefore no longer relevant. This explains our many relatively brief answers below, for which we apologize, but rest assured that all comments have been considered and/or directly included in the new manuscript. At the same time, we want to thank the reviewers for their thorough and constructive comments that helped us to greatly improve our manuscript and the robustness of the results.

- A. Compared to *NCOMMS-20-33873-T*, the focus of the new manuscript moved from the identification/timing of an irreversible "tipping point" in refreezing to the introduction of a new concept, "peak refreezing", and its identification for the Greenland ice sheet. After having reached "peak refreezing", the Greenland ice sheet experiences a permanent refreezing decline that drastically increases meltwater runoff, and therewith the ice sheet contribution to global sea level rise. This change of focus resolves a concern raised by the reviewer about the "irreversibility" of the refreezing decline in the longer term (centuries to millennia).
- B. The new manuscript includes 16 additional climate simulations extending our refreezing time series from 1850 to 2300, whereas the previous manuscript only covered the period 1950-2100. Extending our time series resolves a concern raised by the reviewers about our previous attempt to extrapolate model results beyond 2100 to identify peak refreezing.
- C. These new time series include one short-term projection from our state-of-the-art regional climate model (RACMO2) under a low-end warming scenario (SSP1-2.6) until 2100, as well as three long-term projections until 2300 from the global climate model CESM2 under a low-end (SSP1-2.6) and two high-end warming scenarios (SSP5-8.5). One of these long-term SSP5-8.5 projections includes an interactive ice sheet and hence accounts for ice dynamics (glacier retreat and thinning in time). These new data sets address three previous concerns raised by the reviewers:
 - (1) Uncertainty in correcting the runoff line altitude derived from the global climate model (CESM2), statistically downscaled to 1 km, under different warming scenarios using only one "benchmark" RACMO2 projection under a high-end warming scenario (SSP5-8.5). The inclusion of a new "benchmark" RACMO2 projection under SSP1-2.6 now ensures that our corrections are valid for other scenarios than SSP5-8.5.
 - (2) Uncertainty about whether a low-end warming scenario could potentially cross peak refreezing after 2100. The inclusion of a long-term SSP1-2.6 projection until 2300,

statistically downscaled to 1 km, shows that no peak refreezing is predicted beyond 2100 under a low-end warming scenario.

- (3) Uncertainty about the impact of ice dynamical processes on the timing of peak refreezing. The inclusion of two long-term SSP5-8.5 projections one excluding, the other one including ice dynamics shows that on the time scales considered here glacier retreat and thinning have no significant impact on the timing of peak refreezing.
- D. Our new estimate of peak refreezing does no longer rely on (non-linear) extrapolations beyond 2100 of the runoff line altitude. The additional long-term data sets now explicitly predict the evolution of the runoff line altitude by 2300 under a low-end and two high-end scenarios. It is important to note that these long-term data sets corroborate our previous hypotheses (i.e., based on extrapolations in the original manuscript) but reduce the uncertainty of the timing of peak refreezing, and its impact on Greenland mass loss.
- E. Finally, the new manuscript now clarifies how relevant uncertainties are estimated.
- (1) Uncertainty in estimating time series of SMB components is discussed in L343-345.
 - (2) Uncertainty in estimating time series of runoff line altitude is discussed in L361-362.
 - (3) Uncertainty in estimating the threshold altitude for which $90 \pm 10\%$ of a sector or ice sheet area produces runoff (> 100 mm w.e. per year) is discussed in L362-368.
 - (4) Uncertainty in estimating the timing of peak refreezing is discussed in L377-382.

Reviewer #1 (Remarks to the Author):

KEY RESULTS

This manuscript presents analysis of 20th, 21st, and 22nd century evolution of surface mass balance across the Greenland Ice Sheet, focusing on the infiltration and refreezing of surface meltwater within firn. Key results include snow, rain, meltwater runoff, and surface meltwater infiltration and refreeze across the GrIS for the 20th (1950-2014) and 21st (2015-2099) centuries. Model results were generated using the Community Earth System Model (CESM2) run for various historical simulations and Shared Socioeconomic Pathway emission scenarios, and dynamically downscaled using the Regional Atmospheric climate model version 2.3p2 (RACMO2.3p2). The modeled evolution of GrIS surface mass balance during the 21st century shows a nonlinear response to climate warming as the ablation zone expands and the firn zone retreats. Authors analyze model output to define the runoff line, the uppermost altitude where surface meltwater is at least 100 mm w.e. yr⁻¹. This runoff line advances upward in elevation nonlinearly as GrIS meltwater runoff increases due to snowfall rates remaining relatively stable (which precludes firn replenishment), while rain and surface meltwater increase, the firn column saturates with meltwater refreeze, and the areal extent of the firn zone shrinks. The authors fit a quadratic to the runoff line altitude increase per time from 1950-2099 according to various climate scenarios. These quadratic fits, applied to GrIS hypsometry, are used to project the timing when more than 90% of the GrIS is below this runoff line altitude. Results for high emission scenarios show that this so-called collapse of firn refreezing capacity happens sooner in the 22nd century (2148) than low emission scenarios. The focus of the paper has substantially changed, see discussion above. The paper now discusses “peak refreezing” rather than a “tipping point” in refreezing (Major changes A). In addition, our conclusions do no longer rely on extrapolations beyond 2100 (Major changes B and D).

VALIDITY

The topic of meltwater refreeze within Greenland Ice Sheet firn is of extreme importance to glaciologists, but I am not sure that the work presented here is sufficiently insightful to warrant the kind of public and media attention garnered by Nature Climate Change. What do glaciologists gain from reading this paper? We deem that the conclusions of our manuscript are significant for a broad readership, including but not limited to glaciologists. We show that, after reaching “peak refreezing”, Greenland mass loss increases over twenty-fold, hence accelerating the ice sheet contribution to global sea-level rise. The results are robust: we draw our conclusions based on 51 long-term climate reconstructions/projections, statistically downscaled to 1 km spatial resolution, based on multiple ensemble simulations (each of which including many members) from the latest state-of-the-art version of the CMIP6 CESM2 model that were thoroughly evaluated using benchmark reanalyses-forced RACMO2 simulations. This represents a unique data set capturing 450 years of Greenland firn refreezing under different warming scenarios (1850-2300), which will become publicly available upon publication.

If the current rate of firn saturation due to refreeze keeps up, then the firn will eventually saturate. That saturation point happens in the 22nd century, sooner (2148) that century for high emission/high melt scenario than low emission/low melt scenarios. The quality of this result is consistent with what experts expect. The specific timing might be useful but the method for determining this timing requires more explicit explanation and physical justification of why it is valid. Why is fitting a curve to firn zone shrinkage 1950-2099 accurate for extrapolating that process further inland where the firn column is presumably colder and thicker? This is now addressed using long-term (until 2300) climate projections in the revised manuscript (Major changes B and D).

Why is this curve, which was forced by 1950-2099 climate, applicable to describe Earth system conditions beyond 2100? This is now addressed using long-term (until 2300) climate projections in the revised manuscript (Major changes B and D).

ORIGINALITY AND SIGNIFICANCE

Originality is the good performance/agreement between model output and observations 1950-2014. But this was already reported in (Noël and others, 2019). Projecting the timing and extent of firn saturation due to meltwater infiltration and refreeze, as well as the insight into mass balance partitioning (snow, rain, runoff) for 1950-2099 under Shared Socioeconomic Pathway emission scenarios is significant, I think, because of the downscaling to 1 km. Other applications of the CESM2 for GrIS projections (Muntjewerf and others, 2020) apparently do not achieve this high resolution nor employ recently improved representation of the Greenland firn layer (Ligtenberg and others, 2018). The fact that we downscale the model output to unprecedented resolution is indeed a major novelty of our analysis. Another novelty is that we include a projection that accounts for changes in ice dynamics (glacier retreat and thinning), and show that the latter processes do not substantially impact the timing of “peak refreezing” for the Greenland ice sheet (Major changes C3). In addition, we present a unique data set capturing 450 years of Greenland firn refreezing under different warming scenarios (1850-2300), which will become publicly available upon publication.

UNCERTAINTY TREATMENT

Uncertainty is never very clearly defined in this paper, but the reader can deduce that the presented uncertainty (line 113) is defined as the range of modelled outcomes. This range arises from the different climate scenarios. It therefore reflects spread in the modeled climate forcing, but not necessarily in ice sheet surface mass balance response. Uncertainty due to missing physical processes in the model is not clearly reported. “Uncertainty” appears first on line 113 without clear definition. Uncertainty shown on Figure 3, 4, and 5 is similarly derived from climate model scenario comparisons. The supplement does not expand on uncertainty treatment. The revised manuscript now clarifies how uncertainties are estimated (Major changes E1-4).

The authors acknowledge a few processes that are not accounted in this modeling exercise (lines 147-148) yet I was left wondering about what magnitude would unresolved processes (e.g. firn aquifers, firn aquifer drainage, elevation-lowering-feedback) alter the reported results. Perhaps previous work could be leveraged to estimate: How much of the modelled firn column could conceivably be occupied by aquifer (Forster and others, 2014)? How much water in the aquifer could drain (Poinar and others, 2017)? Irreducible water retention in firn aquifers is accounted for in both the original CESM2 and RACMO2 simulations, which is transferred to the statistically downscaled products at 1 km. When “peak refreezing” is reached, the remaining firn zone that could potentially retain water in aquifers will become smaller and smaller. Therefore, uncertainties in water retention in aquifers will become less and less important. The new manuscript also includes a long-term projection that accounts for changes in ice dynamics (glacier retreat and thinning), and shows that the latter processes do not substantially impact the timing of “peak refreezing” for the Greenland ice sheet (Major changes C3).

How much extra surface melt could be generated were the GrIS model adjusted to show ice sheet thinning and subsequent surface elevation lowering (Muntjewerf and others, 2020)? The new manuscript accounts for ice dynamics and its impact on the timing of peak refreezing (Major changes C3).

CONCLUSIONS

I find the conclusions questionable. For example, Line 166 “Greenland surface mass loss increases non-linearly with the advance of the runoff line, irrespective of the warming scenario” This conclusion, as applied to projections past 2099, is based on a nonlinear quadratic fit to 1950-2099 output then used to project the advance of the runoff line. Therefore, does it necessarily reflect real world future evolution of the GrIS firn meltwater storage capacity? Rather, the conclusion could simply be an artifact of the methodology and the assumption baked into that approach, namely that the saturation and runoff line advance will continue to march upward 2099 onward at the same nonlinear rate modeled for 1950-2099. Why is that assumption valid? The new manuscript includes long-term projections until 2300 and does no longer rely on extrapolations beyond 2100 (Major changes B and D). It is important to note that our previous results are corroborated by the new runs.

LIMITATIONS

I do not work with climate or surface mass balance models on a regular basis and therefore might be missing some common knowledge shared within that community of practice. To

make the work more readily accessible to a wider readership, I suggest that the following technical details regarding the model be presented outright:

-vertical height of the firn column represented in the model, i.e. 40m depth? 100m+ depth? This is now addressed in the new manuscript in L241-243 for CESM2 and L254-256 for RACMO2.

-spatial variability in the vertical height of the firn column represented in the model, i.e. how does this vary across the ice sheet and does model show the firn thinning as well as losing areal extent? Thinning and retreat of the firn layer is indeed accounted for in the snow module of both RACMO2 and CESM2. A detailed description of the RACMO2 and CESM2 snow modules can be found in Ligtenberg et al. (2011, 2018) and Van Kampenhout et al. (2020).

-agreement or disagreement between the chosen regional climate model (RACMO2.3p2) with other regional climate models (HIRHAM5, MARv3.9). Intercomparison of regional climate models is not the purpose of our manuscript. Such exercise has been performed earlier in e.g. Fettweis et al. (2020). RACMO2 shows very good agreement with both in situ and satellite measurements, and compares well with other regional climate models, including HIRHAM and MAR (e.g. Mankoff et al., 2021).

-explain how the statistically downscaled refreezing (line 254), which is calculated as a residual derived from surface melt, rainfall, and runoff, compares to refreezing calculated within firn/snow models thermodynamically constrained (i.e. output prior to downscaling). Such comparison has been previously performed in Noël et al. (2020a) using different RACMO2 products (see their Tables 1 and 2). In general, statistical downscaling enhances melt, runoff and thus refreezing. The resulting downscaled SMB shows excellent agreement with in situ (see Fig. 1c in Noël et al. (2020a)) and satellite measurements (see Fig. 3c in Noël et al. (2020a)). Runoff was also evaluated using discharge measurements from the Watson River in western Greenland, showing very good agreement (see Fig. S4b in Noël et al. (2019)). Another exhaustive study compared RACMO2 runoff with discharge measurements from multiple catchments in Greenland, again showing very good agreement (Mankoff et al. (2020)). Based on these results, and since direct measurements of refreezing do not exist, we deem that refreezing and retention in firn is well captured in the products used here.

- similarly, how does the downscaled refreeze residual (equation 6) compare to independent, direct model output of refreeze? Some validation, and perhaps comparison with other firn models (Stevens and others, 2020), might elucidate the robustness of this Refreezing (RF) estimate. Refreezing estimates at 1 km result from statistically downscaling melt, rainfall and runoff, that are derived from the thoroughly evaluated CESM2 and RACMO2 models. Evaluations of CESM2 and RACMO2 can be found in e.g. Van Kampenhout et al. (2020) and Noël et al. (2019, 2020). In addition, RACMO2 data have been previously used to force different firn models including e.g. IMAU-FDM (Ligtenberg et al., 2018), the Community Firn Model (CFM, Verjans et al., 2019) or SNOWPACK (Steger et al., 2017). All these models produced refreezing outputs that are in good agreement with RACMO2. Firn properties are also in good agreement with observations of density and temperature (Brils et al., 2021). Finally, comparing the downscaled refreezing from RACMO2-CESM2 under SSP1-2.6 or SSP5-8.5 with refreezing from the corresponding original CESM2 projections, we again find good

agreement (see Supplementary Figure 7f). The same holds for other SMB components (Fig. 7a-e). See also our response to the previous comment.

LINE COMMENTS

Title. Consider editing the title to include the word “Projected” to reflect methodology and honor uncertainty associated with forecasts: “Projected 22nd century tipping point in refreezing accelerates Greenland ice sheet mass loss” The title now changed to: “Peak refreezing in the Greenland firn layer under future warming scenarios”.

Line 2. Instead of “collapse of firn refreezing” consider referring to it as “firn saturation” or something that evokes the physical process more directly. Reaching the saturation capacity of the firn is less something collapsing/tumbling/falling/degrading and more a process of saturation/freezing/solidifying/squeezing out pore space. The focus of the paper has substantially changed. The paper now discusses “peak refreezing” rather than a “tipping point” in or a “collapse” of refreezing (Major changes A).

Line 3. Consider stating which Earth System Model outright in the abstract. This is not relevant anymore in the new manuscript.

Line 6. Consider “project” instead of “predict” This is not relevant anymore in the new manuscript.

Line 6. Consider “could reach saturation” instead of “could collapse” This is not relevant anymore in the new manuscript.

Line 11. Consider “likely avoids firn saturation altogether” instead of “likely prevents” This is not relevant anymore in the new manuscript.

Line 16-17. Consider editing to “These retention mechanisms prevent meltwater runoff, mitigating surface mass loss.” This is not relevant anymore in the new manuscript.

Line 28. Consider “In a warming climate, exceeding firn saturation and refreezing capacity is inevitable” instead of “collapse of firn refreezing” This is not relevant anymore in the new manuscript (Major changes A).

Line 64. Consider adding some brief explanation of how RACMO2.3p2 compares to other regional climate models and why it was chosen for this work. Intercomparison of regional climate models is not the purpose of this manuscript. Such an exercise has been performed earlier in Fettweis et al. (2020) and Mankoff et al. (2021). RACMO2 shows good performance compared to in situ and satellite measurements, and compares well with other models including HIRHAM and MAR.

Line 65. Consider changing “we predict” to “model output show the firn zone retreating xx% from current (2020) extent to cover 61%” This is not relevant anymore in the new manuscript.

Line 68. As it is written, the non-linear SMB decline could be explained by the expansion of ablation zone and bare ice where there is zero firn meltwater storage. Consider editing to fit firn zone into this explanation. This is not relevant anymore in the new manuscript.

Line 73. Consider specifying “the remainder of the precipitation input is discharged”. This is not relevant anymore in the new manuscript

Line 84. Consider justifying why this quadratic fit to model output is physically reasonable for the extrapolation. This is not relevant anymore in the new manuscript (Major changes B and D).

Line 87. Consider changing the title to something that speaks to physical process, i.e. “Firn saturation coupled with insufficient snowfall yield tipping points” This is not relevant anymore in the new manuscript (Major changes A).

Line 91. Consider editing to “In the percolation zone, which is the zone extending just above the equilibrium line altitude where some surface melt occurs, refreezing significantly declines” This definition agrees with Cuffey and Paterson (2010) and is more descriptive than “low accumulation zone” This is not relevant anymore in the new manuscript.

Line 100. Consider highlighting this insight in the abstract. Results show that expansion of the firn saturation zone is due not only to meltwater refreeze but also lack of snow to replenish firn. This is not relevant anymore in the new manuscript.

Line 127. Consider editing to “Equation 1 can be extrapolated to predict because X” Questions: How does equation 1 adequately capture 22nd century nonlinearities when it was not trained on 21st century data? Would another inflection point be expected that would call for another, new, different set of coefficients? Alternatively, would it take longer (slower RLA hike) in 22nd century for thicker, colder firn column to saturate progressively inland? The equation is not part of the new manuscript anymore, and the methodology has been substantially changed (Major changes B-D).

Line 162. Consider adding a sentence to elaborate on how this study compares to citation 23, a study where the GrIS is dynamically coupled to CESM (i.e. surface elevation adjusted). The new manuscript accounts for ice dynamics and its impact on the timing of peak refreezing (Major changes C3).

Line 166. “advance of the runoff line” evokes march toward the terminus. Consider instead “hike” or “increase of the runoff line” This is not relevant anymore in the new manuscript.

Line 198. How deep does the 40-layer snow module extend? How does this compare to the observed firn column depth? This is now addressed in L255: “... and includes a 40-layer snow module (up to 100 m depth) simulating melt ...”

Line 292. Correct date on citation Poinar et al. (2017) Done.

Reviewer #2 (Remarks to the Author):

In ‘22nd century tipping point in refreezing accelerates Greenland ice sheet mass loss’, Noël et al. seek to investigate the limits of the buffering effect from meltwater refreezing in the Greenland Ice Sheet’s percolation zone on the overall ice sheet surface mass balance. The authors achieve this by interrogating and downscaling a variety of climate model products, both directly from a CESM (Community Earth System Model), and from a regional climate model (RACMO) simulation, forced by CESM output through the 21st century. Through this process, the authors find that under all emissions scenarios, refreezing on the Greenland Ice Sheet (GrIS) increases through the 21st century as surface melt increases in magnitude in the percolation zone and also occurs at higher elevations. Under the high emissions (SSP5-8.5) scenario, model outputs indicate that the runoff limit increasingly extends inland through the 21st century. Using this as a reasonable proxy for refreezing, extrapolation of the trend implies that refreezing is likely to decline some time in the mid-22nd century. This marks a tipping point for this region of the ice sheet, whereby surface melt is partitioned more and more in to runoff (cf. refreezing), and contributes to sea level rise.

Refreezing in the GrIS’ percolation zone is central to understanding the fate of meltwater, and with it the contributions to sea level rise. The numbers demonstrate the importance: nearly one half of ice sheet melt refreezes in place. The paper topic is therefore highly relevant to a critical issue in projecting ice sheet mass balance. The concept of a ‘tipping point’ is not particularly new, as the authors have illustrated the process in an earlier paper focused on

Greenland's glaciers and ice caps (Noël and others, 2017). But the GrIS tipping point has not yet been investigated. I appreciate the effort by the authors to leverage a number of climate products; their results are not particularly skewed to a single simulation. The importance of the subject and use of a suite of model results are all reasons that the manuscript warrants publication. However, I remain quite skeptical of the results, which I believe are highly speculative for a few reasons:

1) The manuscript's core take-homes rely on extrapolation of model results out some 50-300 years. The new manuscript includes long-term projections until 2300 and does not rely on extrapolations beyond 2100 (Major changes B and D).

2) Critical uncertainties in the buffering capacity of the percolation zone persist, but these are largely related to shortcomings in our understanding of the infiltration process. How impermeable are 'ice slabs'? How deep does meltwater infiltrate, and how do we parameterize complex fingering flow in models that cannot treat such local & heterogeneous processes? Not only are these uncertainties not addressed or overcome in this paper, but the opposite is true -- the firn model in the CESM runs is even more simplistic than the tipping bucket model implemented in RACMO. The representation of (impermeable) ice slabs is, to our knowledge, not included in current regional climate models that typically run at 5 to 11 km spatial resolution. One reason is that these processes are currently poorly understood; another one is that ice slabs are likely not spatially continuous/homogeneous over the area of a model grid cell, i.e., 25 or 121 km². This means that meltwater will eventually find a way to percolate deeper underneath the ice layer, e.g. through piping or a break in the ice slab. In a previous study, Verjans et al. (2019) used the Community Firn Model (CFM) to show that more complex percolation schemes than the commonly used bucket method did not improve the representation of meltwater retention/refreezing in firn. They show that most of the uncertainties originate from the input forcing rather than the percolation method used. Both RACMO2 and CESM2 have been specifically improved to better represent firn processes over polar ice sheets, and were thoroughly evaluated in terms of SMB in previous studies, see e.g. Noël et al. (2019, 2020) and Van Kampenhout (2020).

3) The CESM output is nominally on a 111 km grid. On this scale, the percolation zone in many regions around the ice sheet is ~1 grid cell. The authors overcome this limitation with statistical downscaling, but the downscaling is performed using historical or present-day data and are assumed to remain static in time. For instance, despite the large increase in mass loss reported, all modeling assumes that the ice sheet does not change in elevation over the next century. This is now addressed by including one long-term projection including ice dynamics (glacier thinning and retreat) under a high-end warming scenario (SSP5-8.5) until 2300. Our results show that accounting for ice dynamics does not significantly impact the timing of peak refreezing (Major changes C3).

Considered in aggregate, it's unclear how these uncertainties influence the results. But I suspect the true timescale for the ice sheet to reach a tipping point has an uncertainty envelope that is much larger than the +/- 15 years the authors report based on the model suite. The primary contribution of the manuscript is in projecting a timescale of this percolation zone 'tipping point'. Since this is the case, I believe the fidelity of the model output is of the utmost importance. Because I remain skeptical of the model output, especially in the

absence of any discussion of the sources of uncertainty and the plausible impact on the interpretation, I cannot recommend the manuscript for publication. The addition of multiple long-term projections until 2300, one including ice dynamical processes, have much improved the robustness of our results (Major changes C1-3). We no longer rely on extrapolations beyond 2100 (Major changes B and D). The revised manuscript now explicitly clarifies how uncertainties are estimated (Major changes E1-4).

I recognize that the model outputs are what they are, and to some degree cannot be improved without advances in understanding of the processes controlling meltwater infiltration in firn. But I believe the paper could still be improved if the authors removed the extrapolation and more appropriately acknowledged and discussed the sources of uncertainty in the model outputs. Of course in doing so, the primary take-homes may be far more uncertain, not quite as confidently impactful, and therefore perhaps less suitable to a high-profile journal. The new manuscript includes long-term projections until 2300 and does not rely on extrapolations beyond 2100 (Major changes B and D).

In light of my recommendation, I have not provided detailed comments on the manuscript. But there were three other details of the paper that I'll comment on, and which I believe would improve a revised paper if fixed. I found that the introduction did not motivate a specific problem which the paper set out to solve or advance. The authors acknowledge 'major uncertainties' (line 26-27) in future meltwater refreezing, but fail to communicate what these uncertainties are. If this study overcomes any of these uncertainties, then outline them in the introduction to give the reader a reason to invest time in the paper. Incidentally, the authors state in the next sentence (line 28) that collapse of firn refreezing is inevitable. If this is the case, the authors are stating that the reader shouldn't be investing time in the paper to begin with because we should already know the main conclusion! The paper has been substantially changed and this comment is not relevant anymore (Major change A).

To me, the word 'collapse' implies a rapid time scale and complete disintegration. But I believe the results show neither of these. The results indicate a tipping point in refreezing in the SW sector by the end of the 21st century, but certainly not a rapid extinction of refreezing. The primary model take-home is based on extrapolation of a proxy for this tipping point. Don't get me wrong, I believe the ice sheet is in a major state of change, and this shouldn't be minimized by cautious scientists. But I interpreted the statements of 'collapse' to be an exaggeration that raised red flags with the results. The focus of the paper has now changed, we discuss the timing of reaching "peak refreezing", which does not imply long-term irreversibility (on the centennial to millennial time scale) (Major changes A).

The authors define capacity as the amount of water that has been retained or refrozen (eqn 7), but I believe a more accurate description of this quantity is 'refreezing fraction'. Capacity is the amount of pore space that is available for meltwater refreezing/retention (e.g. see methods in Harper et al., (2012)) and does not depend directly on melt amounts. When the authors describe/present a reduction in refreezing capacity (e.g. Figure 1), this should be termed refreezing fraction. The capacity would be the air pore space integrated over the ice sheet. To remain consistent with previous publications e.g. Noël et al. (2017) and Noël et al. (2020b) published earlier in Nature Communications, we decided to use the term "refreezing

capacity". However, we understand the reviewer's comment, and we agree to use the term "refreezing fraction" in a revised version, if preferred by the editor.

Reviewer #3 (Remarks to the Author):

What are the major claims of the paper?

The manuscript presents simulations from an Earth System Model, CESM2, and a Regional Climate Model, RACMO2.3p2, over Greenland during 1950-2100 and under various Shared Socioeconomic Pathways (SSP) scenarios. The study describes on the evolution of refreezing, a key component of the Greenland ice sheet mass balance. A "tipping point" in meltwater refreezing is simulated in ~2080 in the Southwest (SW) region in scenarios SSP5-8.5 and is concurrent to a runoff area reaching ~90% of that region in 2080. Runoff line altitude is thereafter used as a proxy to describe when runoff area reaches 90% of a given region and when meltwater refreezing is expected to reach this tipping point. Eventually, an extrapolation method is used to predict when this tipping point is reached in different regions and for the ice sheet as a whole beyond 2100, when no simulation is available. The focus of the paper has substantially changed. The paper now discusses "peak refreezing" rather than a "tipping point" in or a "collapse" of refreezing (Major changes A). In addition, our conclusions do not rely on extrapolations beyond 2100 (Major changes B and D).

Are the claims novel? If not, please identify the major papers that compromise novelty

It is the first time, to my knowledge, that the SMB components from SSP-forced CESM2 simulations are being downscaled to 1 km and adjusted for biases. This makes the presented model outputs very robust over the 1950-2100 period with both high spatial resolution and good match with available observations. The tipping point in refreezing in SW during the 21st century is also novel to my knowledge and relies on robust simulations. The extrapolation of future runoff line altitude, runoff area and crossing of a tipping point in refreezing are also new. The revised manuscript extends the refreezing time series at 1 km to 1850-2300 and our conclusions do not rely on extrapolations beyond 2100 anymore (Major changes B-D).

Will the paper be of interest to others in the field? Will the paper influence thinking in the field? Yes. Thanks.

Are the claims convincing? If not, what further evidence is needed?

Despite the value of the presented model runs (product for all SSP, high spatial resolution, adjusted to observations during the historical period...), I have some limitations about the main result of the study: the prediction of a year in the 22nd century when meltwater refreezing reaches a tipping point and collapses. There are two problems with this claim: its wording and its robustness. For the wording, I have an issue with the use of two words: "tipping point" and "collapse".

- Dictionaries define "tipping point" as "the point at which an object is no longer balanced and adding a small amount of weight can cause it to topple". It carries both the idea of imbalance and irreversibility. Here, the phenomenon that is described as a tipping point is the refreezing total reaching a maximum and then starting to decline (Figure 2b). Nothing is said about how irreversible that curve inflexion is. If late efforts are made for climate mitigation, would the snow and firn layer be able to recover and refreezing start to increase again? This is not mentioned in the manuscript and therefore I find the term "tipping point" used without the

appropriate justification. Dictionaries define "Collapse" by "to fall down suddenly or completely". Here the "tipping point" in refreezing that estimated either for the GrIS or for its subregions characterizes the moment when meltwater refreezing start to decrease. This is defined from what is simulated in the SW region in ~2080. But looking at Figure 2b, it is very subjective how to describe the slope inversion that occurs for the SSP5-8.5 scenario, in 2080 and in the SW region. There is indeed a decline of refreezing, but this decline is neither complete (there are still large amounts of meltwater being refrozen in 2099) nor more abrupt than over other periods of the 21st century (see temporary decreases of refreezing in 2045 and 2060 in the CESM2-RACMO run, Figure 2b). The choice of word "collapse" should therefore be better justified. **The focus of the paper has substantially changed. The paper now discusses "peak refreezing" rather than a "tipping point" in or a "collapse" of refreezing (Major changes A).**

The second major issue is relative to the very optimistic uncertainty bounds provided with the year of expected crossing of the tipping point. At the moment, the uncertainty bounds are calculated (as I understand it) from the spread with which SSP scenarios within each ensemble cross the critical runoff line altitude. This uncertainty estimation does not account for 1) the limitations of the assumptions being used and 2) the uncertainty of the extrapolation method. **Our conclusions do not rely on extrapolations beyond 2100 anymore (Major changes B and D). The revised manuscript now explicitly clarifies how uncertainties are estimated (Major changes E1-4).**

- Regarding the solidity of the assumptions being used. First the tipping point is defined as the year when refreezing initiates a long-lasting decline (Figure 2b), then this definition is translated into the year when 90% of a region experiences runoff (l. 120-122), eventually it is translated into the year when the runoff line altitude reaches a regionspecific critical altitude (Figure 4). The study currently fails to provide an estimation of uncertainty at each conceptual leap. For another region than SW, what is the chance to have its refreezing decreasing before the 90% of its area suffers runoff? How is the uncertainty on this 90%, +/-10%, is estimated and how is it used in within the uncertainty estimation of the year of tipping point? What is the chance for the hypsometric relationship used to relate runoff area and runoff altitude, to be the same in the 22nd century as the one used if Figure 4? **We hope that the revised manuscript clarifies the above concerns, see our Major changes A-E.**

- Regarding the extrapolation method, what is the confidence envelope that applies to the quadratic functions presented? This is usually provided by curve-fitting algorithm as uncertainty bounds on the polynomial's coefficients. Even more uncertain: why would the temporal evolution of runoff line altitude be quadratic? What would happen if a linear function, exponential function or polynomials of higher degree were used instead? The study currently does not address this uncertainty on the future trajectory of the runoff line altitude. Considering these two sources of uncertainty, I find it very optimistic to present the year when the "Greenland firn refreezing capacity could collapse" with an uncertainty of "±15" years. **We hope that the revised manuscript clarifies the above concerns. Our conclusions do not rely on extrapolations beyond 2100 anymore (Major changes B and D). The revised manuscript now explicitly clarifies how uncertainties are estimated (Major changes E1-4).**

Are there other experiments that would strengthen the paper further? How much would they improve it, and how difficult are they likely to be?

Better description and quantification of the uncertainty sources and investigation of multiple extrapolation functions could strengthen the manuscript. The authors should describe better how the decrease of meltwater refreezing they see in the SouthWest region qualifies as tipping point or collapse. Our conclusions do not rely on extrapolations beyond 2100 anymore (Major changes B and D). The revised manuscript now explicitly clarifies how uncertainties are estimated (Major changes E1-4).

Are the claims appropriately discussed in the context of previous literature?

Something I am curious about is whether this maximum refreezing in SW Greenland was also seen in previous versions or RACMO such as van Angelen et al. (2013). van Angelen et al. (2013) became the reference model study for firn pore space loss and threatened firn refreezing capacity in Greenland. It could be beneficial to the study to specify how these better models and climate scenarios improved the predictions for meltwater refreezing in Greenland for the 21st century. Additionally, a similar tipping point was described by the authors in Canada and Greenland's glaciers and ice caps (Noël et al., 2017, 2018). Maybe they could be discussed a bit more to highlight the similarities and differences with the Greenland ice sheet. They could also be used to build trust in the chain of assumptions linking the crossing of a tipping point in refreezing to the runoff line reaching a certain critical altitude. Indeed, the single SW region is currently used as basis for describing the tipping point in meltwater refreezing and the SW case is extrapolated in other regions and for the entire ice sheet. Do Greenlandic and Canadian glaciers and ice caps show a similar relationship between tipping point in refreezing, area experiencing runoff and runoff line reaching a critical altitude? These Greenlandic and Canadian glaciers and ice caps can be used to assess the robustness of the approach and better quantify its uncertainty. This is just a suggestion and is not necessary if a robust uncertainty assessment can be presented from the SW region only. We deem that the above suggestions are not relevant anymore given the extension of our time series beyond 2100 that clearly show when "peak refreezing" is reached in all sectors (Fig. 3a-g) and for the whole Greenland ice sheet (Fig. 5a). In addition, estimates based on the evolution of the runoff line altitude in each sector (Supplementary Fig. 3a-h) support and further corroborate the timing of "peak refreezing" found in Figs. 3 and 5. Including the new long-term projections made our results and conclusions more robust.

Is the manuscript clearly written? Yes. Thanks.

Could the manuscript be shortened to aid communication of the most important findings?

No, it is already straight to the point. Thanks.

Have the authors done themselves justice without overselling their claims?

I think a bit more caution with regards to the wording and to the uncertainty estimation would definitely make the work more robust. We improved our description of uncertainties (Major changes E).

Have they been fair in their treatment of previous literature? See the two minor points above. See our response to the two points above.

Have they provided sufficient methodological detail that the experiments could be reproduced? Yes. Thanks.

Is the statistical analysis of the data sound? The analysis is sound apart from the uncertainty estimation in the extrapolation. Our conclusions do not rely on extrapolations beyond 2100 anymore (Major changes B and D). The revised manuscript now explicitly clarifies how uncertainties are estimated (Major changes E1-4).

Should the authors be asked to provide further data or methodological information to help others replicate their work? Yes. If still necessary, please, let us know how we can further improve this.

As a concluding remark, I would like to highlight the quality and the value of the model outputs presented over the 1950-2099 period. The extrapolation into the 22nd century is an interesting preview of what may happen in the future but is rather uncertain to be the main result highlighted by the study (and its title). Potentially, refocusing the study on the tipping point simulated in SW Greenland, and keep the extrapolation as an interesting discussion point, could be a way to make the study more robust. The paper has been substantially changed and this comment is not relevant anymore (Major change A-E).

I. 1: "90%", in line 52 it is 92%. Done.

I. 2: "the firm covering..." Is the ELA considered equivalent to the firm line? This is neglecting the superimposed ice area. It should be mentioned. Alternatively, the sentence could be rephrased to state that 92% of the ice sheet is accumulation area (Fig.1a) and that a majority of that is firm. The paper has been substantially changed and this comment is not relevant anymore.

I.67-71: Consider cutting that sentence in two. The paper has been substantially changed and this comment is not relevant anymore.

I. 74-79 (related to I.257-259): The refreezing capacity is usually defined as the maximum amount of water that can be refrozen within the firm (i.e. the refreezing that the firm is "capable" of). Here it is defined as what is actually refrozen divided by the sum of two factors, melt and rainfall, which are two external factors to the firm. Maybe it should be called refreezing ratio, fraction or index? To remain consistent with previous publications e.g. Noël et al. (2017) and Noël et al. (2020b) published earlier in Nature Communications, we decided to use the term "refreezing capacity". However, we understand the reviewer's comment, and we agree to use the term "refreezing fraction" in a revised version, if preferred by the editor.

I. 77: The point above becomes problematic: Since your refreezing capacity is currently calculated from refreezing which is calculated (in the downscaled product) from runoff (Eq. 6 and 7), it is the increased runoff that causes the decrease of your calculated refreezing capacity, not the other way around. If the refreezing capacity was defined from pore space and cold content, then the sentence would be accurate, but that would not fit with equation (5) anymore. The paper has been substantially changed and this comment is not relevant anymore.

I.85-86: Specify that the time in Eq. 1 is in years. Consider changing "x" to "t" or "yr" (optional). Considering the importance of equation 1 (along with the coefficients in Table S2) in the determination of the tipping point, I would expect to see a better illustration of these extrapolation functions in the main text. A possibility would be to bring Figure 5a earlier, make

the historical runoff line altitude and fitted function more visible (no need to display after 2200?) and bring Table S2 in the main text. The new figure could also display as shaded area the ensemble spread (current uncertainty estimation) but also other types of uncertainties such as the confidence envelope of the quadratic fit or other fitting functions that could equally fit the historical data. **Our conclusions do not rely on extrapolations beyond 2100 anymore (Major changes B and D), and this comment is no more relevant.**

I. 101: “predicted to cross...” Although it was mentioned in the sentence just before, please mention in the same sentence that this result is linked to one or all scenario (e.g. “for these scenarios” or “for SSP5-8.5”). **The paper has been substantially changed and this comment is not relevant anymore.**

I. 109: Same as above, please mention that this is only for the SSP5-8.5 scenarios. Please check the manuscript throughout so that scenario-specific results are always accompanied with a mention of the scenario they relate to. **The paper has been substantially changed and this comment is not relevant anymore.**

I. 116: How is the uncertainty margin of 10% determined? **This is now explained in L361-368 in the section “Runoff line altitude”. See also our Major changes E3.**

I. 117: “1984 +/- 118 m a.s.l., Fig. 3b” please add that this is the runoff line altitude for the period 2080-2099 (?) **Done.**

I. 129-133: I believe that these results should come with a brief reminder of the assumptions they rely on. Something like “under the conditions that the temporal dynamic of runoff lines remains the same into the 22nd century”. These assumptions should also be discussed in more details elsewhere. **The paper has been substantially changed and this comment is not relevant anymore.**

I. 152: Please mention the uncertainty on the climate evolution in the 22nd century, the uncertainty applying to the coefficients of the quadratic function used for extrapolation and the uncertainty applying to the shape of that extrapolation function. **The paper has been substantially changed and this comment is not relevant anymore.**

I.167-171: This extrapolation is also daring. Using a quadratic function fitted to SMB values in the range [-800, 400] Gt yr⁻¹ and extrapolating it to 4655 Gt yr⁻¹ is rather uncertain. What is the confidence envelope that come with this fit? What would other functions (piecewise linear, exponential, polynomials of other degree) give? I am not sure the manuscript gains much from this analysis. **The new manuscript now includes two long-term projections under SSP5-8.5 until 2300, for which peak refreezing is reached in year 2126 ± 14. In new Fig. 5e, it is clear that SMB follows the cubic curve estimated in the period 1850-2100 (Fig. 5d), and the mass loss obtained when peak refreezing is passed is further supported by the two long-term simulations (until 2300).**

I. 175: “a CMIP6 model” maybe rephrase “a model member of CMIP6” or alike. **The paper has been substantially changed and this comment is not relevant anymore.**

I. 189: “despite” makes the “higher climate sensitivity” sound like a bad behavior. Is it? Could you detail a bit more (if it is relevant)? **CESM2 is one of the CMIP6 climate model that has the highest climate sensitivity (temperature increase for a doubling in CO₂), this is not per se a “bad behavior” of the model but this information is worth being noted. Note also that CESM2 is able to realistically represent the present-day climate and SMB of Greenland (see e.g. Noël et al., 2020). In L202-204, we also added: “In climate models with lower climate sensitivity, we expect that the dates of reaching the refreezing peak could be delayed but would occur at a similar level of warming.”.**

I. 188: I only screened reference 23 and thought that it presented only SMB and not its components. If the SMB components were presented before can you make clear how the outputs presented here are different from the ones presented in Muntjewerf et al. (2020). Note that this entry should be updated in the reference section. The paper has been substantially changed and this comment is not relevant anymore.

I.192: "one SSP5-8.5 (2015-2099) CESM2 member". Please specify which member for reproducibility. It should be also explained whether all members within SSP5-8.5 are equivalent (just alternative scenarios) or if within that ensemble they range from lower warming (SSP5?) to stronger warming (SSP8.5?). In the second case, it is even more important to state which member was used to force RACMO2. In the new manuscript we used one historical member (out of the 12 available CESM2 members; 1950-2014), one climate projection under SSP5-8.5 (out of the 7 available CESM2 members; 2015-2099), and one climate projection under SSP1-2.6 (out of the 6 available CESM2 members; 2015-2099). All members of each scenario were run in a similar fashion, and do not deviate much, see e.g. the colored bands in Figs. 3 a-g and 5 a-c representing the minimum/maximum values (bands) from all members used. The colored solid lines represent the ensemble mean.

I.220: I am a bit puzzled by the "the downscaling procedure corrects ... for ... biases". Is the downscaling procedure aiming to reduce the biases of a certain model output with regard to a certain dataset considered more reliable (that is what is done later with precipitation)? or is it just an adjustment method that keeps constant the amount of melt seen in a coarse cell, but within that cell allocates more melt to lower elevation and less melt in higher elevation? In the second case, I am not sure we can talk about "correction" or "bias" since no reference dataset is being used. In the first case, the dataset that is used as reference for the correction should be specified. The downscaling procedure corrects SMB components (except precipitation), i.e., notably melt and runoff, for negative biases (underestimate) on the relatively coarse RACMO2 (5.5 or 11 km) and coarse CESM2 (~111 km) grids, which show smooth surface topography compared to the high-resolution GIMP DEM at 1 km. The downscaling procedure first corrects melt and runoff for these elevation biases using daily (RACMO2) or monthly (CESM2) elevation gradients of melt and runoff estimated on the original model grid. Those gradients are bi-linearly interpolated onto the high-resolution 1 km grid, and applied to the 1 km topography of the GIMP DEM to estimate corrected (statistically downscaled) fields at 1 km. In a second step, the downscaling technique applies a bare ice albedo correction for melt as bare ice albedo is generally overestimated (too bright) on the original smooth model grids. To that end, we use a high-resolution MODIS albedo product at 1 km, averaged for the period 2000-2015, to estimate the amount of melt that was missed as a result of overestimated bare ice albedo in the original models. Once done, the downscaled runoff is adjusted using the fraction of the additional melt that will effectively run-off. This is done by estimating a scaling factor (\$F_{scale}\$ in Noël et al. (2016)) that allows for a best fit between downscaled SMB and in situ observations, i.e., the scaling factor generally increases runoff in the low ablation zone where the coarse original models do not resolve the high mass loss rates. For detailed information about the downscaling technique, we refer the reviewer to our previous work published in Noël et al. (2016).

As mentioned above, precipitation is not statistically downscaled in RACMO2 and CESM2. However, for CESM2, we performed a precipitation adjustment. The reason is that CESM2 experiences a positive accumulation bias (too wet) due to the smooth topography prescribed on the ~111 km grid, i.e., especially in the southeast where precipitation peaks. In the revised

manuscript, we extensively justify this adjustment and explain its implementation in L294-314.

I. 220-221 “individual SMB components (except precipitation), i.e. primarily melt and runoff”, in the next line you add sublimation, maybe replace by a simpler listing of the downscaled variables. The paper has been substantially changed and this comment is not relevant anymore.

I. 231: So the precipitation and snowfall in CESM (but not in CESM2-forced RACMO) are being adjusted to match the reanalysis-forced RACMO output. As I understand, the melt and runoff are being downscaled using topography and MODIS albedo. Can you explain why you don't use topography to adjust the precipitation in CESM or why you don't adjust melt and runoff in CESM2 to match reanalysis-forced RACMO just like is done for precipitation? We apply an adjustment to precipitation in CESM2 because it shows a clear positive bias (too wet) due to the smooth topography on the coarse ~111 km grid. As a result, the distribution of precipitation in CESM2 is not well represented. Applying statistical downscaling on unadjusted CESM2 precipitation would reflect this bias as well on the 1 km grid. Since downscaled melt and runoff do not show significant bias in CESM2 (see Supplementary Fig. 7), no additional adjustment was required. For more information, we refer the reviewer to the CESM2 evaluation presented in Van Kampenhout et al. (2020). See also our previous response in L220.

I.252-253: “reanalysis-forced RACMO2” In Figure S4b, the caption states that it is the CESM2-forced RACMO2.3p2 output (not reanalysis-forced). Which one is used? . The paper has been substantially changed and this comment is not relevant anymore.

I.257 and 279: Do I understand right that the CESM2 output is adjusted to the reanalysis-forced RACMO2 output first in the downscaling step and then a second time when the SMB components are integrated over specific regions? Is the second adjustment necessary? Sorry for the confusion, the downscaled CESM2 components are not adjusted using RACMO2 data. The reason is that after applying statistical downscaling to CESM2, we obtain excellent agreement with the corresponding downscaled RACMO2 product (see Supplementary Fig. 7). So no further adjustment was required.

It is legitimate to use the historical part of CESM2-forced RACMO2 output to adjust the CESM2 SMB components and runoff line altitude. But it should be clear throughout the manuscript when it is the CESM2 output that is being used or a series that was adjusted to CESM2-forced RACMO2. It also adds to my discussion point on uncertainties: how would the extrapolation functions look on the non-adjusted runoff line altitude time series? We clarified these points in the Methods section ‘Runoff line altitude’ in L355-361. We now use historical RACMO2-CESM2 data to adjust the runoff line altitude of the downscaled CESM2 industrial (IND, 1850-1949) and historical (HIST, 1950-2014) simulations; and a combination of all our RACMO2-CESM2 products (including historical data and the two projections under SSP1-2.6 and SSP5-8.5) to adjust the runoff line altitude in the downscaled CESM2 projections under different warming scenarios. See also our **Major changes C1**.

Figure 1cd: It is hard for me to see the green line. We tried to improve the display in the new Fig. 1.

Figure 2: In the caption, at the end of the 7th line it should be “d Refreezing...”. In the following line: “CESM2-forced RACMO2.3p2 projection”, please add which SSP scenario is used. The figures and captions have been substantially changed and this comment is not relevant anymore.

Figure 3:

- Color of the minimal firn area: Can you use a color that is not already part of the runoff colormap? We tried to improve visibility of these contours in the new Fig. 4.

- The use of "tipping point" in the legend: It seems here that several quantities go into "tipping point": A year when refreezing start to decline (Fig 2), an area that experiences runoff (Figure 4) or a "critical runoff line altitude" (Fig.3). I think the legend here should show "critical runoff line" and the caption should tell briefly why that line is "critical" and how it relates to the tipping point in refreezing. The figures have been substantially changed and this comment is not relevant anymore.

- "In a-f, the blue contour represents the critical runoff line altitude and associated uncertainty (dashed blue) marking the GrIS-wide tipping point in firn refreezing." I see the dashed blue only in (a), is that normal? I also find it hard to distinguish these dashed lines from the elevation contour. The figures have been substantially changed and this comment is not relevant anymore.

- "The cyan area outlines the remaining firn area after the tipping point is passed." Please be more specific: In which scenario? In which year? If there are still some firn remaining, how does it qualify as a collapse or tipping point? The cyan line outlines the \$90 \pm 10\%\$ area contour, for which peak refreezing is reached, i.e., \$90 \pm 10\%\$ of the area produces runoff \$> 100\$ mm w.e. per year. This threshold is similar for all scenarios.

Figure 5: How is calculated the uncertainty that applies to the year when CESM2-forced RACMO simulation reaches the critical runoff line altitude (black dot with whiskers)? I have understood that the uncertainty in the crossing of this threshold was derived from the spread seen within each ensemble. For the CESM-forced RACMO2 simulation, there is only one member being used (l.192). The paper has been substantially changed and this comment is not relevant anymore.

Additional references:

1. Ligtenberg, S. R. M., Helsen, M. M., and van den Broeke, M. R.: An improved semi-empirical model for the densification of Antarctic firn, *The Cryosphere*, 5, 809–819, <https://doi.org/10.5194/tc-5-809-2011>, 2011.
2. Ligtenberg, S. R. M., Kuipers Munneke, P., Noël, B. P. Y., and van den Broeke, M. R.: Brief communication: Improved simulation of the present-day Greenland firn layer (1960–2016), *The Cryosphere*, 12, 1643–1649, <https://doi.org/10.5194/tc-12-1643-2018>, 2018.
3. Van Kampenhout, L., Lenaerts, J. T. M., Lipscomb, W. H., Lhermitte, S., Noël, B., Vizcaino, M., et al. (2020). Present-day Greenland Ice Sheet climate and surface mass balance in CESM2. *Journal of Geophysical Research: Earth Surface*, 125, e2019JF005318. <https://doi.org/10.1029/2019JF005318>
4. Fettweis, X., Hofer, S., Krebs-Kanzow, U., Amory, C., Aoki, T., Berends, C. J., Born, A., Box, J. E., Delhasse, A., Fujita, K., Gierz, P., Goelzer, H., Hanna, E., Hashimoto, A., Huybrechts, P., Kapsch, M.-L., King, M. D., Kittel, C., Lang, C., Langen, P. L., Lenaerts, J. T. M., Liston, G. E., Lohmann, G., Mernild, S. H., Mikolajewicz, U., Modali, K., Mottram, R. H., Niwano, M., Noël, B., Ryan, J. C., Smith, A., Streffing, J., Tedesco, M., van de Berg, W. J., van den Broeke, M., van de Wal, R. S. W., van Kampenhout, L., Wilton, D., Wouters, B., Ziemen, F., and Zolles, T.: GrSMBMIP: intercomparison of the modelled 1980–2012 surface mass balance over the Greenland Ice Sheet, *The Cryosphere*, 14, 3935–3958, <https://doi.org/10.5194/tc-14-3935-2020>, 2020.

5. Mankoff, K. D., Solgaard, A., Colgan, W., Ahlstrøm, A. P., Khan, S. A., and Fausto, R. S.: Greenland Ice Sheet solid ice discharge from 1986 through March 2020, *Earth Syst. Sci. Data*, 12, 1367–1383, <https://doi.org/10.5194/essd-12-1367-2020>, 2020.
6. Mankoff, K. D., Fettweis, X., Langen, P. L., Stendel, M., Kjeldsen, K. K., Karlsson, N. B., Noël, B., van den Broeke, M. R., Solgaard, A., Colgan, W., Box, J. E., Simonsen, S. B., King, M. D., Ahlstrøm, A. P., Andersen, S. B., and Fausto, R. S.: Greenland ice sheet mass balance from 1840 through next week, *Earth Syst. Sci. Data*, 13, 5001–5025, <https://doi.org/10.5194/essd-13-5001-2021>, 2021.
7. Noël, B., van de Berg, W. J., Machguth, H., Lhermitte, S., Howat, I., Fettweis, X., and van den Broeke, M. R.: A daily, 1 km resolution data set of downscaled Greenland ice sheet surface mass balance (1958–2015), *The Cryosphere*, 10, 2361–2377, <https://doi.org/10.5194/tc-10-2361-2016>, 2016.
8. Noël, B., van de Berg, W., Lhermitte, S. et al. A tipping point in refreezing accelerates mass loss of Greenland's glaciers and ice caps. *Nat Commun* 8, 14730 (2017). <https://doi.org/10.1038/ncomms14730>
9. B. Noël, W. J. van de Berg, S. Lhermitte, M. R. van den Broeke, Rapid ablation zone expansion amplifies north Greenland mass loss. *Sci. Adv.*5, eaaw0123 (2019).
10. Noël, B., van Kampenhout, L., van de Berg, W. J., Lenaerts, J. T. M., Wouters, B., and van den Broeke, M. R.: Brief communication: CESM2 climate forcing (1950–2014) yields realistic Greenland ice sheet surface mass balance, *The Cryosphere*, 14, 1425–1435, <https://doi.org/10.5194/tc-14-1425-2020>, 2020a.
11. Noël, B., Jakobs, C.L., van Pelt, W.J.J. et al. Low elevation of Svalbard glaciers drives high mass loss variability. *Nat Commun* 11, 4597 (2020b). <https://doi.org/10.1038/s41467-020-18356-1>
12. Verjans, V., Leeson, A. A., Stevens, C. M., MacFerrin, M., Noël, B., and van den Broeke, M. R.: Development of physically based liquid water schemes for Greenland firn-densification models, *The Cryosphere*, 13, 1819–1842, <https://doi.org/10.5194/tc-13-1819-2019>, 2019.
13. Steger CR, Reijmer CH, van den Broeke MR, Wever N, Forster RR, Koenig LS, Kuipers Munneke P, Lehning M, Lhermitte S, Ligtenberg SRM, Miège C and Noël BPY (2017) Firn Meltwater Retention on the Greenland Ice Sheet: A Model Comparison. *Front. Earth Sci.* 5:3. doi: 10.3389/feart.2017.00003
14. Brils, M., Kuipers Munneke, P., van de Berg, W. J., and van den Broeke, M.: Improved representation of the contemporary Greenland ice sheet firn layer by IMAU-FDM v1.2G, *Geosci. Model Dev. Discuss.* [preprint], <https://doi.org/10.5194/gmd-2021-303>, in review, 2021.

REVIEWER COMMENTS

Reviewer #5 (Remarks to the Author):

Please see attached file for comments.

Review of “Peak Refreezing in the Greenland Firn Layer Under Future Warming Scenarios”

This paper uses a series of 51 global and regional climate model reconstructions and projections of Greenland surface mass balance to project the date at which peak meltwater refreezing occurs in the Greenland firn. Four categories of model results are considered – climate projections covering 1950-2099 from Community Earth System model (CESM2) dynamically downscaled using the regional climate model RACMO2.3p2, a series of pre-industrial and historical CESM2 runs covering 1850-2014, an ensemble of statistically downscaled CESM2 projections covering 2014-2099, and three statistically downscaled CESM2 projections covering 2014-2300. Projections encompass the full range of Shared Socioeconomic Pathway emissions scenarios. The authors particularly analyze the firn refreezing capacity – defined as the percentage of rain and meltwater refrozen in the firn – in these time series at both the IMBIE basin and ice sheet scales. They find that under SSP5-8.5 and over the longest timescales, the ice sheet sees a peak and decline in firn refreezing capacity that occurs first in the SW sector but is reached across the whole ice sheet by around 2200. Based on the behavior of SW Greenland, the authors suggest that peak refreezing occurs when runoff is produced across 90% of the ice sheet and they use hypsometric relationships to predict this timing across other sectors of the ice sheet and for the SSP3-7.0 scenarios which do not extend beyond 2099. The long-term model runs suggest that once peak refreezing is reached, Greenland’s contribution to sea level rise will increase 23-fold relative to the 1992-2018 period due to an ongoing decline in firn storage capacity leading to increase surface meltwater runoff.

To my knowledge, this is the first paper to produce high-resolution (1km) projections of Greenland SMB components over this long of a time scale and over such a broad range of future climate scenarios. The identification of a timescale for Greenland to reach peak firn refreezing and the long-term trajectory of Greenland SMB component partitioning under this declining firn pack is also new. Given the large ensemble of projections from two models that have been fairly well validated for use in the polar regions, I think that the authors have been generally done a responsible job designing the study given the current state of polar climate and firn models. However, given the large firn model uncertainties that are hard to quantify (for example, the impact of missing processes like firn aquifers, ice slab formation, and subsurface lateral flow), it is not clear to me how meaningful the projected dates of peak refreezing can be, although the general understanding of the shape of the response and how it differs under different SSP scenarios is still a useful baseline to have in the literature.

Major Comments:

[1] I am very confused about the hypsometry exercise and when the reported timescales and runoff areas are taken directly from the models versus being estimated from the hypsometric relationship. Given that the results include three fully spatially-resolved models run out to 2300 that encompass the SSP5-8.5 projected peak in refreezing, I do not fully understand the purpose of using the hypsometric scaling. After digging through the methods, it seems that maybe this allows the authors to correct for biases in CESM2 total refreezing relative to RACMO2.3p2 before predicting the timing of peak refreezing and therefore it is used to reduce bias in the results, but that is only my conjecture. And if it is the case, that puts a huge caveat on the time series shown in Figures 3 and 5, since it suggests they are not particularly useful on their own.

I think the manuscript would be significantly strengthened with more clarification (in the main text, not buried in the methods!!) of when, where, and why the hypsometry relationship was used, how it was developed, and what biases are implied in the full model results. Specific questions it would be helpful to have addressed:

- Why is peak refreezing timing set based on the extrapolation of the runoff altitude to the 90% point, rather than just by looking at the refreezing total in the models that run out to 2300 (as shown in Figure 3)?
- Why don't the timing estimates from the adjusted runoff altitude match the apparent refreezing peaks in the actual model output for all sectors? Is this because of the CESM bias adjustment? Or is the 90% threshold not a good one in those sectors?
- How was the 90% runoff area threshold chosen? Is this based only on models run from the southwest up to 2099 as the manuscript seems to imply? If so, why, given that you have model runs out to 2300 that cover the whole ice sheet?
- Exactly which numbers in the paper come from the full model runs versus hypsometric scaling and why?

[2] The methods state the firn module within CESM only models the upper 10 meters of the firn pack. The implications of this setup need significantly more discussion given that this manuscript is trying to model the extinction of a firn pack that is currently up to ~100m thick in the Greenland interior. I agree that field measurements suggest that meltwater does not tend to infiltrate much deeper than 10m, so in a relatively steady state firn pack, this could be acceptable. But as you allow the climate to warm and regions reach an annual melt to accumulation ratio greater than 1, older firn will start to move back up into the top 10m. It seems like you could end up in a problematic situation where winter accumulation pushes the bottom layers out of the resolved depth, and summer melt should move those layers back into it, but the model no longer has information about those older layers.

[3] Statistical downscaling makes sense for producing a reliable high-resolution product at any given timestep, but since the firnpack is evolving dynamically, biases in precipitation on the native model grid seem like they would lead to large biases in firnpack thickness that will impact storage capacity. I understand that runoff and melt also get corrected, but since this is happening offline, it seems like model biases in precipitation and melt are going to dynamically propagate through the treatment of the firnpack and potentially skew runoff numbers in the interior in unpredictable ways.

[4] According to Supplementary Figure 7, it seems that CESM2 significantly overestimates refreezing relative to RACMO2.3p2. This is a very important caveat to the long-term CESM2-only model results that are highlighted in the main text, particularly when it comes to the timing of peak refreezing. How is this bias incorporated into those estimates (if at all)? It would be very helpful to have that point clarified in the main text somewhere.

[5] What about firn aquifers in the Southeast? This manuscript projects that the Southeast will hit peak refreezing almost as quickly as the Southwest, even though these sectors currently show wildly different responses to meltwater input, with ice slab formation in the SW and firn aquifer maintenance in the SE. This section of the results would be greatly strengthened with more discussion of how the models treat firn aquifers and whether they can reliably simulate the long-term evolution of a largely temperate and saturated firn pack. Essentially, it is not clear to me what is going to drive the current Helheim firn aquifer to transition into almost a bare ice zone in the next 80 years, which is what these models seem to be projecting.

Minor Comments:

Supplementary Figure 2 – why does this not show the model runs out to 2300? What is the point of this figure given that you have those long-term runs that could validate your hypsometric relationship?

Why was the period 2080-2099 chosen for all of the trends?

Consider providing (in the methods of supplement) a list of which CMIP6 ensemble members were used in this paper. In the main text, it would be helpful to provide a short sentence describing what is different about ensemble members within the same SSP scenario. Different boundary conditions, different model parameters, etc?

Line 80 - It would be good to briefly (one sentence) summarize the findings of Noel (2020). How good is the agreement with observations over this time period and are there any key biases to be aware of? Is there good agreement at the basin level or just in aggregate for the ice sheet?

Line 84 – “We let the historical SMB time series transition into two projections...” It is not clear what this means. Are the historical reconstructions run forward into projections in the model, or are you just saying that there is a transition point on the plots between different model runs?

Line 86 – consider citing the IMBIE basin division for your seven sectors.

Line 104 – the definition of refreezing capacity is confusing. I would tend to think of refreezing capacity as total storage space (like firn air content) and the ratio as defined in this paper as something like “melt retention fraction”.

Line 164 – Is it really all that useful to provide an ice-sheet averaged elevation for the runoff line when it can vary so widely between different sectors? I assume this is a non-unique average in that you could have a wide variety of different runoff altitude scenarios that might average out to this number but would not necessarily lead to the same total runoff area (given the different hypsometry of various sectors).

Line 168-171 – I am confused by the use of “predict” versus “project” in these sentences. Is this supposed to be a subtle indicator that some of these numbers come directly from the models and others from the hypsometric scaling?

Line 174 – Why cite Figure 4 here? It is not clear to me how Figure 4 supports this statement.

Line 192 – Lower bound on what? The time at which peak refreezing occurs? The runoff line elevation? The amount of total warming? If you mean the timing of peak refreezing, would your projections not more likely be an upper bound, since all of the processes you list in 1-3 would lead to more melt or more rapid depletion of firn storage capacity?

Line 195 – How much of an issue is algal and bacterial darkening in the firn zone? I’m familiar with these albedo-melt feedbacks increasing melt in the bare ice zone, but that would not really impact the rate of firn air depletion at higher elevations.

Line 357 – Is uncertainty in the regression coefficients propagated into the uncertainty estimate? If not, why not? Particularly in this case where the relationships do not seem entirely linear (although I think the

linear fit choice is reasonable since there is not a distinctly more appropriate model and there is not reason to introduce unneeded complexity).

Reviewer #6 (Remarks to the Author):

The manuscript by Noël et al., uses a robust set of high resolution climate simulations to explore the firn layer's response—namely the refreezing capacity—to future warming scenarios. The main finding is that under the most extreme warming scenario (SSP5-8.5) parts of the Greenland Ice Sheet can experience peak refreezing by the end of the century, with the remainder of the ice sheet closely following in the early 2100's. Peak refreezing, or refreezing within the firn layer, is important because it acts to store meltwater, preventing direct runoff. This work shows that the refreezing capacity of the ice sheet's firn layer will decrease as the climate warms and more melting occurs further inland at higher elevations. The authors present robust and sound results and conclusions, that demonstrate the importance of the firn layer as a buffer to sea level rise and show how the refreezing capacity of the firn is not static but will evolve in response to climatic warming. The results of this work are therefore both relevant to the glaciological community and beyond. Ultimately, this work makes a compelling argument that firn's refreezing capacity is a critical physical process that needs to be incorporated into future projections of the Greenland Ice Sheet's sea level rise projections.

The authors appear to have addressed all of the main concerns described during the first round of review, with the current manuscript now including additional simulations, uncertainties, and more clearly stating the work's significance.

The authors were successful in reworking the manuscript to focus on the new concept of "peak refreezing". All major questions I had while reading the main text were addressed and answered by the methods or by the supplementary figures. While minor, I strongly recommend moving the definitions of some key concepts to earlier in the manuscript. For example, peak refreezing isn't defined until line 153 (toward the end of the results section), I highly recommend moving this up to the first mention of the concept within the Results section and taking the time therein to define this central concept (similar suggestion applies to the runoff line altitude). Other than the minor comments detailed below (some of which detail some figure readability issues), the only other point of concern was the last line of the text (Line 214). I question if the warming scenery represented by SSP5-8.5 is "unlikely to be met in the future". The way this sentence is written seems presumptuous, I would recommend rephrasing to something like "only under the most extreme warming scenarios will peak refreezing be reached, highlighting the resilience of Greenland firn...". Finally, if space allows, the manuscript would benefit from more explanation on why some areas will experience peak refreezing before others. From the main text and supplemental figures it appears that sector area (and maximum elevation) are important factors along with the projected runoff line altitude increase over time. An elaboration on controls for the rate over which the runoff line altitude increases for different sectors would be interesting and would answer the question of why peak refreezing will be reached in the south earlier than elsewhere.

Minor Comments

Lines 85-86: The "seven sectors" correspond to ice divides or drainage basins. Consider adding the citation to Rignot, E., & Mouginot, J. (2012). Ice flow in Greenland for the international polar year 2008–2009. *Geophysical Research Letters*, 39(11).

Line 191: change to "prevent the occurrence of GrIS-wide peak refreezing."

Line 192: remove "note that" and start sentence with "Our estimates..."

Figures

A general and very minor comment is that the light blue line color that appears in Figures 1-2 is not cyan according to how the color appears in the included PDF (colors may have been converted to another colormap upon converting document to pdf).

Consider removing the black outline from the Greenland map's bedrock boundaries (leaving those areas gray instead), this will help to increase visibility of contours and of the ice sheet's extent.

Figure 1

The gray color corresponding to SMB-ERA is very hard to see in subplots c and d, even when significantly enlarging the figure. I suggest changing this color to increase readability.

Figure 2

a - It is very difficult to see the light blue solid line, consider closing a darker blue for better contrast against the dark red line.

A-c - the dashed lines are difficult to see as well. Consider using a different shade and/or increase dash separation

Figure 3 and 4

Consider using a perceptually uniform colormap for runoff. The current colormap (what appears to be JET) is not color-blind friendly, is not readable if printed in black and white.

Figure 5

Mislabel sub-label c as b in caption. Also, the sea level rise scale on subplot d is important but with so much going on it hard to find/see. Maybe consider putting those values in bold. Removing some of the sea level rise tick mark labels in subplot e would declutter the scale bar as well.

Supplementary Figure 2 - Great figure

J. Mejia

Reviewer #7 (Remarks to the Author):

This is a revised manuscript that has already gone through a round of revisions based on the initial reviewers' comments and concerns. The authors' response to reviewers appears to be thorough and addresses all the initial comments. Due to an extensive revision of the manuscript, some concerning parts of the previous manuscript are no longer present. Some of the main concerns of the original manuscript were the extrapolation beyond the model-run years for when the "tipping point" would be reached, the use of the term "tipping point", and a lack of description of the uncertainties in the methods. All of these appear to have been thoroughly addressed in this revised version of the manuscript by extending the time series, changing the terminology and focus to "peak refreezing" instead of "tipping point", and including descriptions of how the uncertainties were estimated.

This manuscript investigates the impact of warming under a range of Shared Socioeconomic Pathway emissions scenarios on the "peak refreezing" of the seven sectors of the Greenland Ice Sheet (GrIS), as well as the GrIS as a whole, by using downscaled CESM2 climate projections to drive the RACMO2.3p2 model for the time period of 1850-2300. The methodology is sound, and now contains an appropriate description of the uncertainties raised by the initial reviewers. The authors present a unique and important dataset by downscaling the model results to a resolution of 1 km. Under the high-warming scenario (SSP5-8.5), the authors find that peak refreezing on the GrIS would occur by 2126 +/- 14 years, when global warming reaches approximately 9.1 +/- 0.9C in the Arctic. Under this scenario, the authors predict that Greenland's contribution to global sea level rise will increase 23-fold to approximately 9.2 +/- 6.3 mm/yr, and that the Southwest and Southeast sectors will reach peak refreezing first. Importantly, the authors find that under lower warming scenarios, refreezing stabilizes, and "peak freezing" does not occur.

These are important results as they demonstrate that even moderate climate efforts to decrease

climate change could allow for the GrIS to retain a significant amount of meltwater storage capacity through the next couple of centuries, buffering sea level rise due to Greenland melt. Therefore, this manuscript should be of interest to the broad readership of Nature Communications. I only have a few comments below that may help to further improve this manuscript:

General Comments:

- I may not be up-to-date on my understanding of projected future snow accumulation rates above the ablation zone in Greenland, but I was under the impression that the snow accumulation rate was expected to increase with warming due to the increased water vapor capacity of warmer air (e.g., Hawley et al., 2017 – Greenland traverse firn core study), and that models like RACMO underestimate current accumulation rates by about 35% on the GrIS (at least in the Northern sectors; e.g., Karlsson et al., 2020). I imagine that a projected increase in accumulation rate wouldn't persist beyond some threshold date, but your low warming scenario shows a slight decrease, and your high warming scenario shows a fairly steady rate of precipitation through the end of the century. Since precipitation is an important factor in SMB estimates, how do you reconcile these discrepancies between observations and model output, with your model results?
- For the low-warming scenario (SSP1-2.6) why does the refreezing capacity appear to increase right after 2100?

Technical Corrections:

Line 145: describing the 1960-1989 runoff line in the past tense will make this sentence less confusing

Figure 1 caption:

- towards the end of "a)" add ", respectively" after "2080-2099" so that the reader knows that the black line and hatching are indicating two different things on the figure. Consider defining the seven sectors in Figure 1a here or somewhere in the text, as this journal reaches a broad audience, and/or indicate in the last sentence of caption 1a that the 7 sectors are shown in Figure 1a with the other black lines on the figure and the acronyms.
- At the end of the first sentence for "c and d" add ", respectively." To clarify that c is showing the SSP1-2.6 scenario and d the SSP5-8.5 scenario.
- In the last sentence, consider adding something that defines this simulation period as "ERA"

Figure 2:

- What do the dashed lines from 2080-2099 in "b" and "c" represent?

Figure 3:

- If you define the sector acronym earlier in the text, it will be easier for the reader to know that plots a-g are aligned in this multi-panel figure generally next to their sector on the central panel

Figure 5 caption:

- For "b" consider rephrasing to something like "Time series of GrIS-integrated anomalies in the upper atmospheric global temperature (500 hPa) relative to pre-industrial (1850-1949). Thick coloured lines represent ensemble mean ... for the same model scenarios described in "a"
- Change the second "b" to "c" and use similar structure as suggestion above to describe panel c

Dear editor and reviewers, we would like to thank you for the thorough comments on our manuscript. Below, you will find our point-by-point answers in blue, with modifications applied to the revised manuscript in red.

Reviewer #5 (Remarks to the Author):

This paper uses a series of 51 global and regional climate model reconstructions and projections of Greenland surface mass balance to project the date at which peak meltwater refreezing occurs in the Greenland firn. Four categories of model results are considered – climate projections covering 1950-2099 from Community Earth System model (CESM2) dynamically downscaled using the regional climate model RACMO2.3p2, a series of pre-industrial and historical CESM2 runs covering 1850-2014, an ensemble of statistically downscaled CESM2 projections covering 2014-2099, and three statistically downscaled CESM2 projections covering 2014-2300. Projections encompass the full range of Shared Socioeconomic Pathway emissions scenarios. The authors particularly analyze the firn refreezing capacity – defined as the percentage of rain and meltwater refrozen in the firn – in these time series at both the IMBIE basin and ice sheet scales. They find that under SSP5-8.5 and over the longest timescales, the ice sheet sees a peak and decline in firn refreezing capacity that occurs first in the SW sector but is reached across the whole ice sheet by around 2200. Based on the behavior of SW Greenland, the authors suggest that peak refreezing occurs when runoff is produced across 90% of the ice sheet and they use hypsometric relationships to predict this timing across other sectors of the ice sheet and for the SSP3-7.0 scenarios which do not extend beyond 2099. The long-term model runs suggest that once peak refreezing is reached, Greenland’s contribution to sea level rise will increase 23-fold relative to the 1992-2018 period due to an ongoing decline in firn storage capacity leading to increase surface meltwater runoff.

To my knowledge, this is the first paper to produce high-resolution (1km) projections of Greenland SMB components over this long of a time scale and over such a broad range of future climate scenarios. The identification of a timescale for Greenland to reach peak firn refreezing and the long-term trajectory of Greenland SMB component partitioning under this declining firn pack is also new. Given the large ensemble of projections from two models that have been fairly well validated for use in the polar regions, I think that the authors have been generally done a responsible job designing the study given the current state of polar climate and firn models. However, given the large firn model uncertainties that are hard to quantify (for example, the impact of missing processes like firn aquifers, ice slab formation, and subsurface lateral flow), it is not clear to me how meaningful the projected dates of peak refreezing can be, although the general understanding of the shape of the response and how it differs under different SSP scenarios is still a useful baseline to have in the literature. Thank you for the positive comments. We acknowledge that uncertainties remain due to firn processes currently not represented in climate models, and we expect these uncertainties to persist, given the difficulty to observe these processes in the field. Yet, we deem that our results are accurate in first order and provide, as the reviewer states, a useful baseline. This is supported by the fact that contemporary refreezing is accurately modelled, as evidenced by the very good agreement between modelled recent GrIS mass loss and GRACE (e.g., Van den Broeke et al., 2016), and between modelled and measured runoff in different catchments across the GrIS (e.g., Mankoff et al., 2020). In line with this, Langen et al. (2017) confirmed that differences in integrated runoff remain small when including ice slabs in the RCM HIRHAM. Future work on representation of ice slabs and aquifers should further refine these estimates. We added the following in the discussion: “Among others, missing processes include the formation of thick, impermeable ice lenses inhibiting active meltwater percolation and retention in firn (MacFerrin et al., 2019), although Langen et al. (2017) previously showed that accounting for these in a regional climate model had little impact on GrIS-integrated runoff.”

Major Comments:

[1] I am very confused about the hypsometry exercise and when the reported timescales and runoff areas are taken directly from the models versus being estimated from the hypsometric relationship. Given that the results include three fully spatially-resolved models run out to 2300 that encompass the SSP5-8.5 projected peak in refreezing, I do not fully understand the purpose of using the hypsometric scaling. After digging through the methods, it seems that maybe this allows the authors to correct for biases in CESM2 total refreezing relative to RACMO2.3p2 before predicting the timing of peak refreezing and therefore it is used to reduce bias in the results, but that is only my conjecture. And if it is the case, that puts a huge caveat on the time series shown in Figures 3 and 5, since it suggests they are not particularly useful on their own. I think the manuscript would be significantly strengthened with more clarification (in the main text, not buried in the methods!!) of when, where, and why the hypsometry relationship was used, how it was developed, and what biases are implied in the full model results. We apologize for the confusion and agree that we could have described more clearly why and when the hypsometry approach is used. The timing and uncertainty of peak refreezing are derived from the regional hypsometry thresholds shown in Supplementary Fig. 3. The main reason for using the hypsometry threshold approach is that we want to estimate the timing of peak refreezing in the SSP3-7.0 scenario, for which all model projections end in 2100. An added advantage of applying this method is that we can estimate timing of peak refreezing based on substantially more projections: 8 SSP5-8.5 members can be used in the hypsometry approach compared to only 2 SSP5-8.5 when using the long-term refreezing time series until 2300. We find that our timing estimates for regional peak refreezing using the hypsometry approach agree very well with the refreezing peak (and following decline) identified in the two long-term SSP5-8.5 scenarios. See our answers to specific questions below.

Specific questions it would be helpful to have addressed:

- Why is peak refreezing timing set based on the extrapolation of the runoff altitude to the 90% point, rather than just by looking at the refreezing total in the models that run out to 2300 (as shown in Figure 3)?
- How was the 90% runoff area threshold chosen? Is this based only on models run from the southwest up to 2099 as the manuscript seems to imply? If so, why, given that you have model runs out to 2300 that cover the whole ice sheet? Here we answer these two questions combined. We clarified by adding to the main text: “To identify whether other sectors and the GrIS as a whole will reach peak refreezing, we complement our two CESM2-forced RACMO2.3p2 projections with 48 downscaled CESM2 simulations (see Methods). These simulations include 25 CESM2 ensemble projections until 2100 (SSP1-2.6, SSP2-4.5, SSP3-7.0 and SSP5-8.5), as well as three long-term CESM2 projections until 2300 under extended SSP1-2.6 and SSP5-8.5 scenarios (see Methods). To estimate the regional timing of peak refreezing, even for projections that do not extend beyond 2100, we extrapolate the runoff line altitude in time, defined as demarcating the region producing over 100 mm w.e. of runoff per year (see Methods). To do this, we use as a benchmark the results from the SW sector where peak refreezing is reached before 2100 in all SSP5-8.5 scenarios (8 members) (Fig. 3d). We find that as firn saturates and retreats in SW Greenland, the runoff line migrates upward in a non-linear fashion under SSP5-8.5 (Supplementary Fig. 3a), from \$1537 \pm 228\$ m a.s.l. (meters above sea level) in 1960-1989 to a threshold altitude of \$2514 \pm 159\$ m a.s.l., when peak refreezing is crossed (Supplementary Fig. 2a). Peak refreezing in SW is reached when \$90 \pm 10\%\$ of the sector area produces runoff, i.e., 90% of the sector area is situated below the runoff line. Applying this \$90 \pm 10\%\$ condition to sector hypsometries, i.e., area-elevation distribution, yields regional thresholds for runoff line altitude (blue lines in Supplementary Fig. 2). The timing of peak refreezing in each sector is then obtained by extrapolating the increasing runoff line altitude until it crosses this threshold value (blue lines in Supplementary Fig. 3).”
- Why don't the timing estimates from the adjusted runoff altitude match the apparent refreezing peaks in the actual model output for all sectors? Is this because of the CESM bias adjustment? Or is the 90% threshold not a good one in those sectors? On the contrary, Figs. 3a-g and 5a show that the timing of peak refreezing in SSP5-8.5 derived from our hypsometry threshold approach (red dots) agrees within uncertainty (red whiskers) with peak refreezing (and subsequent decline) identified in the actual model (red lines). We added for clarification to the text: “Figures 3a-g and 5a show that the timing of peak refreezing in SSP5-8.5 obtained from extrapolation (red dots) agrees within uncertainty (red whiskers) with the peak identified in the two long-term refreezing projections under SSP5-8.5 (red lines), confirming the robustness of our method.”
- Exactly which numbers in the paper come from the full model runs versus hypsometric scaling and why? Only the timing of peak refreezing and its uncertainty (red [resp. orange] dots and whiskers for SSP5-8.5 [resp. SSP3-7.0] in Figs. 3 and 5) are derived from the hypsometry threshold approach.

[2] The methods state the firn module within CESM only models the upper 10 meters of the firn pack. The implications of this setup need significantly more discussion given that this manuscript is trying to model the extinction of a firn pack that is currently up to ~100m thick in the Greenland interior. I agree that field measurements suggest that meltwater does not tend to infiltrate much deeper than 10m, so in a relatively steady state firn pack, this could be acceptable. But as you allow the climate to warm and regions reach an annual melt to accumulation ratio greater than 1, older firn will start to move back up into the top 10m. It seems like you could end up in a problematic situation where winter accumulation pushes the bottom layers out of the resolved depth, and summer melt should move those layers back into it, but the model no longer has information about those older layers. This is unlikely to cause problems as we find very good agreement (\$R^2 = 0.88\$ ) between refreezing from the dynamically downscaled RACMO2.3p2-CESM2 products (with deep firn layer) and the corresponding native CESM2 products (with shallow firn layer, see Supplementary Fig. 7). Difference in modelled refreezing is small (28 Gt or 4.5%) and not statistically significant as it is smaller than 2 standard deviations (\$2SD = 67\$ Gt for 1950-2099). Furthermore, refreezing from our benchmark RACMO2.3p2-ERA product agrees well with the RACMO2.3p2-CESM2 products (black line in Supplementary Fig. 1) and the native CESM2 data both on the regional (black lines in Fig. 3 a-g) and GrIS-wide scales (black line in Fig. 5a). To point to the small differences, the revised Supplementary Fig. 7 now lists two additional statistics, i.e., the average model difference (RACMO2.3p2-CESM2 minus native CESM2) both as an absolute (Gt) and relative (% of change) value. This is now clarified in the figure caption as: “Listed are number of years (N), slope (a) and intercept (b) of the linear regression, coordination coefficient (\$R^2\$ ), and model difference (RACMO2.3p2 minus CESM2) expressed as an absolute (\$\Delta\$ ) and relative (\$\Delta\%\$ ) value.”

[3] Statistical downscaling makes sense for producing a reliable high-resolution product at any given timestep, but since the firnpack is evolving dynamically, biases in precipitation on the native model grid seem like they would lead to large biases in firnpack thickness that will impact storage capacity. I understand that runoff and melt also get corrected, but since this is happening offline, it seems like model biases in precipitation and melt are going to dynamically propagate through the treatment of the firnpack and potentially skew runoff numbers in the interior in unpredictable ways. This is unlikely to cause problems as we find very good agreement between total precipitation (\$R^2 = 0.98\$, \$\Delta = 3\$ Gt or 0.4%), melt (\$R^2 = 0.97\$, \$\Delta = 36\$ Gt or 5.2%) and runoff (\$R^2 = 0.97\$, \$\Delta = 31\$ Gt or 5.1%) from the dynamically downscaled RACMO2.3p2-CESM2 products and the corresponding native CESM2 products (see Supplementary Fig. 7). None of these differences are statistically significant: \$2SD = 29\$ Gt, 138 Gt and 120 Gt for total precipitation, melt and runoff respectively. In the revised manuscript we added a sentence to the Methods: “We find strong correlations (\$R^2 = 0.88 - 0.97\$ ) in GrIS-integrated SMB and components between the CESM2-forced RACMO2.3p2 projections (1950-2099) and the corresponding CESM2 historical (1950-2014) and SSP1-2.6/SSP5-8.5 forcing (2015-2099) (Supplementary Fig. 7). These high correlations also hold for the different sectors of the

GrIS (Supplementary Fig. 8). Model differences between GrIS-integrated SMB and components are also small. Notably, CESM2 has lower melt (36 Gt or 5.2%) and runoff (31 Gt or 5.1%), combined with larger rainfall (33 Gt or 37%), resulting in larger refreezing (28 Gt or 4.5%). Note that the larger relative difference in rainfall originates from the mass flux being small compared to other components. Nevertheless, none of these model differences are statistically significant, i.e., differences are smaller than two standard deviations, highlighting the good agreement between the two downscaling methods.” For data beyond 2100, we can only rely on the 3 long-term CESM2 scenarios (two SSP5-8.5 and one SSP1-2.6) which do not deviate from the short-term projections (2015-2099), providing an almost seamless transition after 2100. In addition, refreezing in the two long-term SSP5-8.5 scenarios do not differ much until 2200 both on the regional scale (Fig. 3a-g) and GrIS-wide scale (Fig. 5a), i.e., well after peak refreezing is crossed for all sectors and the entire GrIS under SSP5-8.5.

[4] According to Supplementary Figure 7, it seems that CESM2 significantly overestimates refreezing relative to RACMO2.3p2. This is a very important caveat to the long-term CESM2-only model results that are highlighted in the main text, particularly when it comes to the timing of peak refreezing. How is this bias incorporated into those estimates (if at all)? It would be very helpful to have that point clarified in the main text somewhere. The refreezing difference between both models is actually small (-28 Gt or -4.5%). It is not corrected as the difference is not statistically significant (2SD = 67 Gt for 1950-2099). The difference partly originates from the fact that native CESM2 produces more rainfall than the dynamically downscaled RACMO2.3p2-CESM2 (Supplementary Fig. 7e). In addition, refreezing from our benchmark RACMO2.3p2-ERA product at 1 km agrees very well with the RACMO2.3p2-CESM2 products at 1 km (black line in Supplementary Fig. 1) and the native CESM2 data at 1 km both on the regional (black lines in Fig. 3 a-g) and GrIS-wide scale (black line in Fig. 5a), highlighting the robustness of our results. See also our previous response.

[5] What about firn aquifers in the Southeast? This manuscript projects that the Southeast will hit peak refreezing almost as quickly as the Southwest, even though these sectors currently show wildly different responses to meltwater input, with ice slab formation in the SW and firn aquifer maintenance in the SE. This section of the results would be greatly strengthened with more discussion of how the models treat firn aquifers and whether they can reliably simulate the long-term evolution of a largely temperate and saturated firn pack. Essentially, it is not clear to me what is going to drive the current Helheim firn aquifer to transition into almost a bare ice zone in the next 80 years, which is what these models seem to be projecting. We agree, and now explicitly state the following in the discussion: “Furthermore, none of the models used here explicitly simulate fully saturated firn aquifers, although RACMO2.3p2 reliably captures the occurrence of perennial liquid water by capillary retention in the contemporary firn (Forster et al., 2014). This means that present-day runoff from the active firn aquifers of SE Greenland is likely overestimated, potentially affecting the future evolution of the aquifers and their contribution to mass loss.”

Minor Comments:

Supplementary Figure 2 – why does this not show the model runs out to 2300? In all sectors and GrIS-wide, the $90 \pm 10\%$ threshold (blue solid line and blue belt) is crossed by mid-2100 under SSP5-8.5, thereafter the runoff line reaches the highest elevation of each sector by 2200. For SSP1-2.6, the runoff line altitude stabilises after 2100. For these reasons, showing data until 2300 in Supplementary Fig. 2 would only overwhelm the reader with non-essential information: overlapping of multiple horizontal lines on the upper bound of the blue belt for SSP5-8.5 projections beyond the mid-2100 since the runoff line altitude reaches the top of the ice sheet; and overlapping of multiple horizontal lines on the already existing SSP1-2.6 result for 2080-2099 since the runoff line altitude stabilises after 2100.

This information is already shown in the time series of Supplementary Fig. 3. We agree that there is some overlap. However, Supplementary Fig. 2 allows the reader to identify that when the runoff line altitude reaches 2514 ± 159 m a.s.l., it represents $90 \pm 10\%$ of the sector area. Such relationship cannot be drawn from Supplementary Fig. 3 alone, making these two figures complementary.

What is the point of this figure given that you have those long-term runs that could validate your hypsometric relationship? Please see our previous responses.

Why was the period 2080-2099 chosen for all of the trends? This period was selected to produce a smooth SMB map in Fig. 1a (i.e. not having to use an individual year), and is used for consistency throughout the paper. It also represents the period in which SW Greenland crosses peak refreezing as identified in Fig. 2c.

Consider providing (in the methods of supplement) a list of which CMIP6 ensemble members were used in this paper. In the main text, it would be helpful to provide a short sentence describing what is different about ensemble members within the same SSP scenario. Different boundary conditions, different model parameters, etc? We added to the Methods section: “These simulations were produced and made available by the National Centre for Atmospheric Research (NCAR). The various SSP scenarios are described in O’Neill et al. (2015); the difference between each member of a same scenario stems from a random perturbation applied to the temperature field used for initialisation.”

Line 80 - It would be good to briefly (one sentence) summarize the findings of Noel (2020). How good is the agreement with observations over this time period and are there any key biases to be aware of? Is there good agreement at the basin level or just in aggregate for the ice sheet? Comparison of model-derived and GRACE mass change shows good agreement as

discussed in the main text. We added new Supplementary Figure 8 below and the following sentence: “Overall, CESM2-forced RACMO2.3p2 shows very good agreement with SMB observations, with only a small bias in both the accumulation (-21 mm w.e.) and ablation zone (180 mm w.e.), of similar quality to the benchmark reanalysis-forced RACMO2.3p2 product (-22 mm w.e. and 120 mm w.e., respectively) (Noël et al., 2020). This agreement is confirmed for GrIS-integrated SMB components (Supplementary Fig. 7), and for SMB on the regional scale (Supplementary Fig. 8).”

Supplementary Figure 8: Cross-model correlation of regional SMB. a-h Correlation of sector integrated SMB between the CESM2-forced RACMO2.3p2 and the corresponding CESM2 products statistically downscaled to 1 km spatial resolution for the historical period (1950-2014) and for both SSP1-2.6 and SSP5-8.5 warming scenarios (2015-2099). Listed are number of years (N), slope (α) and intercept (β) of the linear regression, coordination coefficient (R^2), and model difference (RACMO2.3p2 minus CESM2) expressed as an absolute value (Δ).

Line 84 – “We let the historical SMB time series transition into two projections...” It is not clear what this means. Are the historical reconstructions run forward into projections in the model, or are you just saying that there is a transition point on the plots between different model runs? Here we mean that the two projections (SSP1-2.6 and SSP5-8.5, 2015-2099) are branched from the same historical run (1950-2014). This is now clarified as: “We extend the historical SMB time series (1950-2014) with two projections for a low-end and high-end emission scenario (2015-2099) (Noël et al., 2021).”

Line 86 – consider citing the IMBIE basin division for your seven sectors. Following reviewer #7, we reformulated the sentence as: “To explore regional changes in firn processes, we divide the GrIS in seven sectors⁷ (Fig. 1a): southwest (SW), southeast (SE), central west (CW), central east (CE), north (NO), northwest (NW) and northeast (NE).” and added a reference to Mougnot et al. (2019), where these seven sectors were first defined.

Line 104 – the definition of refreezing capacity is confusing. I would tend to think of refreezing capacity as total storage space (like firn air content) and the ratio as defined in this paper as something like “melt retention fraction”. We understand the reviewer’s comment but would prefer keeping the terminology “refreezing capacity” to remain consistent with our previous publications e.g. Noël et al. (2017) and Noël et al. (2020). We are happy to replace “refreezing capacity” by “refreezing fraction” at the discretion of the editor.

Line 164 – Is it really all that useful to provide an ice-sheet averaged elevation for the runoff line when it can vary so widely between different sectors? I assume this is a non-unique average in that you could have a wide variety of different runoff altitude scenarios that might average out to this number but would not necessarily lead to the same total runoff area (given the different hypsometry of various sectors). For consistency with the analysis done for individual sector, we deem that this information should remain in the main text. Especially since this threshold runoff line altitude was used to time GrIS-wide peak refreezing.

Line 168-171 – I am confused by the use of “predict” versus “project” in these sentences. Is this supposed to be a subtle indicator that some of these numbers come directly from the models and others from the hypsometric scaling? There is no subtle meaning here, we used ‘predict’ and ‘project’ interchangeably to avoid unnecessary repetitions. We reformulated as: “Figures 3a-g and 5a show that the timing of peak refreezing in SSP5-8.5 obtained from extrapolation (red dots) agrees within uncertainty (red whiskers) with the peak identified in the two long-term refreezing projections under SSP5-8.5 (red lines), confirming the robustness of our method.”

Line 174 – Why cite Figure 4 here? It is not clear to me how Figure 4 supports this statement. We agree that it is not intuitively clear and removed the reference to Fig. 4e-f.

Line 192 – Lower bound on what? The time at which peak refreezing occurs? The runoff line elevation? The amount of total warming? If you mean the timing of peak refreezing, would your projections not more likely be an upper bound, since all of the processes you list in 1-3 would lead to more melt or more rapid depletion of firn storage capacity? Thank you for pointing this out, we corrected as: “Our estimates of the timing of (regional) peak refreezing should be considered an upper bound since additional ...”

Line 195 – How much of an issue is algal and bacterial darkening in the firn zone? I’m familiar with these albedo-melt feedbacks increasing melt in the bare ice zone, but that would not really impact the rate of firn air depletion at higher elevations. We agree that this effect may be marginal in the firn zone and removed the sentence.

Line 357 – Is uncertainty in the regression coefficients propagated into the uncertainty estimate? If not, why not? Particularly in this case where the relationships do not seem entirely linear (although I think the linear fit choice is reasonable since there is not a distinctly more appropriate model and there is not reason to introduce unneeded complexity). The reason why the correlations between native CESM2 and RACMO2.3p2-CESM2 are not linear, notably in the early period of the HIST and SSP runs, is due to the coarse resolution of the native CESM2 data, i.e., one pixel experiencing runoff can cover an area of ~10,000 km² spreading the runoff inland to higher elevations. As a result, the runoff line altitude is overestimated in the early period, an issue that becomes less important as climate warms and runoff propagates towards the ice sheet interior. To address this, we decided to estimate linear regression coefficients (slope and intercept) based on a least-square fit. These coefficients are then used to adjust the runoff line altitude found for the coarser resolution native CESM2 products. As the reviewer states we deemed this the best approach possible to refine our time series of runoff line altitude. Because the adjustment required is more important for the early period (IND and HIST) we used regression coefficients based on the HIST data only, whereas for future projections, the adjustment required is less important and regression coefficients were estimated using the full data set (HIST + SSPs). We clarified this as follows in the revised text: “The correlation between CESM2 and RACMO2.3p2 can differ in the historical and scenario periods (Supplementary Fig. 9). This results from the coarse resolution of CESM2 that propagates runoff too far inland the ice sheet, leading to an overestimated runoff line altitude in the historical period. This resolution artefact becomes less pronounced as runoff is produced further inland following atmospheric warming in the scenario projections. To address this, we estimate linear regression coefficients, i.e., regression ...”

Reviewer #6 (Remarks to the Author):

The manuscript by Noël et al., uses a robust set of high resolution climate simulations to explore the firn layer's response—namely the refreezing capacity—to future warming scenarios. The main finding is that under the most extreme warming scenario (SSP5-8.5) parts of the Greenland Ice Sheet can experience peak refreezing by the end of the century, with the remainder of the ice sheet closely following in the early 2100's. Peak refreezing, or refreezing within the firn layer, is important because it acts to store meltwater, preventing direct runoff. This work shows that the refreezing capacity of the ice sheet's firn layer will decrease as the climate warms and more melting occurs further inland at higher elevations. The authors present robust and sound results and conclusions, that demonstrate the importance of the firn layer as a buffer to sea level rise and show how the refreezing capacity of the firn is not static but will evolve in response to climatic warming. The results of this work are therefore both relevant to the glaciological community and beyond. Ultimately, this work makes a compelling argument that firn's refreezing capacity is a critical physical process that needs to be incorporated into future projections of the Greenland Ice Sheet's sea level rise projections. The authors appear to have addressed all of the main concerns described during the first round of review, with the current manuscript now including additional simulations, uncertainties, and more clearly stating the work's significance. The authors were successful in reworking the manuscript to focus on the new concept of "peak refreezing". All major questions I had while reading the main text were addressed and answered by the methods or by the supplementary figures. Thank you very much!

While minor, I strongly recommend moving the definitions of some key concepts to earlier in the manuscript. For example, peak refreezing isn't defined until line 153 (toward the end of the results section), I highly recommend moving this up to the first mention of the concept within the Results section and taking the time therein to define this central concept (similar suggestion applies to the runoff line altitude). We agree that peak refreezing and runoff line altitude should be defined earlier in the main text. We addressed this by adding the following sentence in the introduction: “In a warming climate, we expect the refreezing of liquid water in the firn to reach a maximum before it irreversibly declines. The timing of this ‘peak refreezing’ is relevant for projections of future GrIS mass loss and the central topic of this paper.” The term runoff line altitude is now defined at its first occurrence in the main text as: “..., defined as demarcating the region producing over 100 mm w.e. of runoff per year (see Methods).”

Other than the minor comments detailed below (some of which detail some figure readability issues), the only other point of concern was the last line of the text (Line 214). I question if the warming scenery represented by SSP5-8.5 is "unlikely to be met in the future". The way this sentence is written seems presumptuous, I would recommend rephrasing to something like “only under the most extreme warming scenarios will peak refreezing be reached, highlighting the resilience of Greenland firn...”. Thank you, we reformulated accordingly: “A $9.1 \pm 0.9^\circ\text{C}$ upper-atmosphere warming is required for GrIS

refreezing to peak, demonstrating that only under the most extreme warming scenarios will peak refreezing be reached. This highlights the resilience of Greenland firn and its refreezing mechanism for the current ice sheet geometry.”

Finally, if space allows, the manuscript would benefit from more explanation on why some areas will experience peak refreezing before others. From the main text and supplemental figures it appears that sector area (and maximum elevation) are important factors along with the projected runoff line altitude increase over time. An elaboration on controls for the rate over which the runoff line altitude increases for different sectors would be interesting and would answer the question of why peak refreezing will be reached in the south earlier than elsewhere. Thank you for this suggestion, we now include the following text in the discussion: “Three parameters control the differences in timing of regional peak refreezing: i) the reference runoff line altitude when the GrIS was in approximate equilibrium in 1960-1989 (Supplementary Table 1), ii) the threshold runoff line altitude to cross peak refreezing (Supplementary Table 1), and iii) the rate of upward migration of the runoff line (Supplementary Fig. 3) that depends on the sector hypsometry (Supplementary Fig. 2) and the rate of regional atmospheric warming. The sectors in the south (SW and SE) pass peak refreezing first: they experience the warmest atmospheric conditions, and have the highest reference and the lowest threshold runoff line altitude. The sectors in the north (NO, NE and NW) have relatively gently sloping ice sheet margins and experience the fastest atmospheric warming in Greenland (Noël et al., 2019), resulting in the fastest upward migration of the runoff line (Supplementary Fig. 3c, d and g). NO crosses peak refreezing before NE and NW due to its lower elevated interior, and thus lower threshold altitude (Supplementary Fig. 2d). Central Greenland (CE and CW) passes peak refreezing the latest as it encompasses the highest regions of the GrIS, resulting in higher threshold runoff line altitudes (Supplementary Fig. 2), while experiencing slower upward migration of the runoff line (Supplementary Fig. 3b and f), equivalent to that of the south (Supplementary Fig. 3a and e).”

Minor Comments

Lines 85-86: The “seven sectors” correspond to ice divides or drainage basins. Consider adding the citation to Rignot, E., & Mouginot, J. (2012). Ice flow in Greenland for the international polar year 2008–2009. *Geophysical Research Letters*, 39(11). Following comments from reviewers #5 and #7, we reformulated as: “To explore regional changes in firn processes, we divide the GrIS in seven sectors⁷ (Fig. 1a): southwest (SW), southeast (SE), central west (CW), central east (CE), north (NO), northwest (NW) and northeast (NE).” and added a reference to Mouginot et al. (2019), where the seven sectors were first defined.

Line 191: change to “prevent the occurrence of GrIS-wide peak refreezing.” Thank you, we reformulated accordingly.

Line 192: remove “note that” and start sentence with “Our estimates...” Based on reviewer #5 comments, we reformulated as: “Our estimates of the timing of (regional) peak refreezing should be considered an upper bound since additional ...”

Figures

A general and very minor comment is that the light blue line color that appears in Figures 1-2 is not cyan according to how the color appears in the included PDF (colors may have been converted to another colormap upon converting document to pdf). Thank you for pointing this, we have now homogenized the colors used in Fig. 1a and Figs. 2a-c to match the colors used in e.g., Fig. 3. The red and cyan belts are now lighter, and we made the regression dashed lines darker to increase their visibility. We hope that our changes improve the overall readability.

Consider removing the black outline from the Greenland map’s bedrock boundaries (leaving those areas gray instead), this will help to increase visibility of contours and of the ice sheet’s extent. Unfortunately, the black outlines are part of our sector and land/sea masks and cannot be removed.

Figure 1 The gray color corresponding to SMB-ERA is very hard to see in subplots c and d, even when significantly enlarging the figure. I suggest changing this color to increase readability. We have now changed the color to green, which we hope is more visible. We modified the caption accordingly.

Figure 2a - It is very difficult to see the light blue solid line, consider closing a darker blue for better contrast against the dark red line. We have now homogenized the colors used in Fig. 1a and Figs. 2a-c to match the colors used in e.g., Fig. 3. The red and cyan belts are now lighter, and we made the dashed regression lines darker to increase their visibility. We hope that our changes improve the overall readability.

A-c - the dashed lines are difficult to see as well. Consider using a different shade and/or increase dash separation. We made the dashed regression lines darker to increase their visibility. We hope that our changes improve the overall readability.

Figure 3 and 4 Consider using a perceptually uniform colormap for runoff. The current colormap (what appears to be JET) is not color-blind friendly, is not readable if printed in black and white. Here we used a similar colormap as in e.g., Fig. 1A in Noël et al. (2019). For consistency with our previous publications, we would prefer keeping the color scale similar. If the editor finds this critical, we are happy to try to update our color scale.

Figure 5 Mislabel sub-label c as b in caption. Done.

Also, the sea level rise scale on subplot d is important but with so much going on it hard to find/see. Maybe consider putting those values in bold. Removing some of the sea level rise tick mark labels in subplot e would declutter the scale bar as well. Indeed, the tickmarks were too small on Fig. 5d, we have now increased their font so that the axes are easier to read. We decided to keep the sea level rise axis as is (tickmark of 2 mm SLR), as we deem that it makes it easier to estimate what change in runoff line elevation results in a 1 mm sea level rise.

Supplementary Figure 2 - Great figure Thank you!

Reviewer #7 (Remarks to the Author):

This is a revised manuscript that has already gone through a round of revisions based on the initial reviewers' comments and concerns. The authors' response to reviewers appears to be thorough and addresses all the initial comments. Due to an extensive revision of the manuscript, some concerning parts of the previous manuscript are no longer present. Some of the main concerns of the original manuscript were the extrapolation beyond the model-run years for when the "tipping point" would be reached, the use of the term "tipping point", and a lack of description of the uncertainties in the methods. All of these appear to have been thoroughly addressed in this revised version of the manuscript by extending the time series, changing the terminology and focus to "peak refreezing" instead of "tipping point", and including descriptions of how the uncertainties were estimated. Thank you.

This manuscript investigates the impact of warming under a range of Shared Socioeconomic Pathway emissions scenarios on the "peak refreezing" of the seven sectors of the Greenland Ice Sheet (GrIS), as well as the GrIS as a whole, by using downscaled CESM2 climate projections to drive the RACMO2.3p2 model for the time period of 1850-2300. The methodology is sound, and now contains an appropriate description of the uncertainties raised by the initial reviewers. The authors present a unique and important dataset by downscaling the model results to a resolution of 1 km. Under the high-warming scenario (SSP5-8.5), the authors find that peak refreezing on the GrIS would occur by 2126 +/- 14 years, when global warming reaches approximately 9.1 +/- 0.9C in the Arctic. Under this scenario, the authors predict that Greenland's contribution to global sea level rise will increase 23-fold to approximately 9.2 +/- 6.3 mm/yr, and that the Southwest and Southeast sectors will reach peak refreezing first. Importantly, the authors find that under lower warming scenarios, refreezing stabilizes, and "peak refreezing" does not occur. These are important results as they demonstrate that even moderate climate efforts to decrease climate change could allow for the GrIS to retain a significant amount of meltwater storage capacity through the next couple of centuries, buffering sea level rise due to Greenland melt. Therefore, this manuscript should be of interest to the broad readership of Nature Communications. I only have a few comments below that may help to further improve this manuscript: Thank you very much!

General Comments:

- I may not be up-to-date on my understanding of projected future snow accumulation rates above the ablation zone in Greenland, but I was under the impression that the snow accumulation rate was expected to increase with warming due to the increased water vapor capacity of warmer air (e.g., Hawley et al., 2017 – Greenland traverse firn core study), and that models like RACMO underestimate current accumulation rates by about 35% on the GrIS (at least in the Northern sectors; e.g., Karlsson et al., 2020). I imagine that a projected increase in accumulation rate wouldn't persist beyond some threshold date, but your low warming scenario shows a slight decrease, and your high warming scenario shows a fairly steady rate of precipitation through the end of the century. Since precipitation is an important factor in SMB estimates, how do you reconcile these discrepancies between observations and model output, with your model results? Unfortunately, the Karlsson study used an obsolete RACMO2.1 SMB dataset (we notified the authors of this omission), the agreement with the data we use here is much improved. We agree that the near-constant accumulation over Greenland is unexpected, but it is a common feature of most projections of Greenland climate, and we have no reason to mistrust this result. We added to the revised manuscript: "Overall, CESM2-forced RACMO2.3p2 shows very good agreement with SMB observations, with only a small bias in both the accumulation (-21 mm w.e.) and ablation zone (180 mm w.e.), of similar quality to the benchmark reanalysis-forced RACMO2.3p2 product (-22 mm w.e. and 120 mm w.e., respectively) (Noël et al., 2020)."
- For the low-warming scenario (SSP1-2.6) why does the refreezing capacity appear to increase right after 2100? First, we want to report that we found a minor bug in the display of refreezing from the long-term SSP1-2.6 data sets in Fig. 3a-g, and Fig. 5a and c. This is now corrected and has no impact on the results and conclusions. The sudden increase in refreezing capacity for SSP1-2.6 after 2100 in Fig. 5c is due to the transition from an ensemble mean of 7 short-term SSP1-2.6 scenarios (2015-2099) to a single long-term member of SSP1-2.6 (2100-2299).

Technical Corrections:

Line 145: describing the 1960-1989 runoff line in the past tense will make this sentence less confusing We reformulated this sentence as: "We find that as firn saturates and retreats in SW Greenland, the runoff line migrates upward in a non-linear fashion under SSP5-8.5 (Supplementary Fig. 3a), from 1537 ± 228 m a.s.l. (meters above sea level) in 1960-1989 to a threshold altitude of 2514 ± 159 m a.s.l., when peak refreezing is crossed (Supplementary Fig. 2a)."

Figure 1 caption: towards the end of “a)” add “, respectively” after “2080-2099” so that the reader knows that the black line and hatching are indicating two different things on the figure. Done.

Consider defining the seven sectors in Figure 1a here or somewhere in the text, as this journal reaches a broad audience, and/or indicate in the last sentence of caption 1a that the 7 sectors are shown in Figure 1a with the other black lines on the figure and the acronyms. We reformulated L85-86 as: “To explore regional changes in firn processes, we divide the GrIS in seven sectors⁷ (Fig. 1a): southwest (SW), southeast (SE), central west (CW), central east (CE), north (NO), northwest (NW) and northeast (NE).” and added a reference to Mouginito et al. (2019), where the seven sectors were first defined.

At the end of the first sentence for “c and d” add “, respectively.” To clarify that c is showing the SSP1-2.6 scenario and d the SSP5-8.5 scenario. Done.

In the last sentence, consider adding something that defines this simulation period as “ERA”
We reformulated the sentence as: “The black line in b and green lines in c-d represent the RACMO2.3p2 simulation forced by ERA reanalyses (1958-2020)¹⁰.”

Figure 2: What do the dashed lines from 2080-2099 in “b” and “c” represent? Thank you for pointing this out, we added the following sentence: “Dashed coloured lines in a-c represent trends for the period 2080-2099.”

Figure 3: If you define the sector acronym earlier in the text, it will be easier for the reader to know that plots a-g are aligned in this multi-panel figure generally next to their sector on the central panel. We reformulated as: “To explore regional changes in firn processes, we divide the GrIS in seven sectors⁷ (Fig. 1a): southwest (SW), southeast (SE), central west (CW), central east (CE), north (NO), northwest (NW) and northeast (NE).” and added a reference to Mouginito et al. (2019), where the seven sectors were first defined.

Figure 5 caption: For “b” consider rephrasing to something like “Time series of GrIS-integrated anomalies in the upper atmospheric global temperature (500 hPa) relative to pre-industrial (1850-1949). Thick coloured lines represent ensemble mean ... for the same model scenarios described in “a” Thank you, we rephrased as: “b Time series of anomalies in the upper atmospheric global temperature (500 hPa) relative to pre-industrial (1850-1949). Thick coloured lines represent ensemble mean temperature anomalies for the same model scenarios described in a.”

Change the second “b” to “c” and use similar structure as suggestion above to describe panel c Thank you, this is corrected.

Additional references:

- Van de Broeke et al. (2016): van den Broeke, M. R., Enderlin, E. M., Howat, I. M., Kuipers Munneke, P., Noël, B. P. Y., van de Berg, W. J., van Meijgaard, E., and Wouters, B.: On the recent contribution of the Greenland ice sheet to sea level change, *The Cryosphere*, 10, 1933–1946, <https://doi.org/10.5194/tc-10-1933-2016>, 2016.
- Mankoff et al. (2020): Mankoff, K. D., Noël, B., Fettweis, X., Ahlstrøm, A. P., Colgan, W., Kondo, K., Langley, K., Sugiyama, S., van As, D., and Fausto, R. S.: Greenland liquid water discharge from 1958 through 2019, *Earth Syst. Sci. Data*, 12, 2811–2841, <https://doi.org/10.5194/essd-12-2811-2020>, 2020.
- Langen et al. (2017): Langen P.L., Fausto R.S., Vandecrux B., Mottram R.H. and Box J.E. (2017) Liquid Water Flow and Retention on the Greenland Ice Sheet in the Regional Climate Model HIRHAM5: Local and Large-Scale Impacts. *Front. Earth Sci.* 4:110. doi: 10.3389/feart.2016.00110
- Noël et al. (2017): Noël, B., van de Berg, W., Lhermitte, S. et al. A tipping point in refreezing accelerates mass loss of Greenland’s glaciers and ice caps. *Nat Commun* 8, 14730 (2017). <https://doi.org/10.1038/ncomms14730>
- Noël et al. (2019): B. Noël, W. J. van de Berg, S. Lhermitte, M. R. van den Broeke, Rapid ablation zone expansion amplifies north Greenland mass loss. *Sci. Adv.* 5, eaaw0123 (2019).
- Noël et al. (2020): Noël, B., Jakobs, C.L., van Pelt, W.J.J. et al. Low elevation of Svalbard glaciers drives high mass loss variability. *Nat Commun* 11, 4597 (2020). <https://doi.org/10.1038/s41467-020-18356-1>

REVIEWERS' COMMENTS

Reviewer #5 (Remarks to the Author):

I would like to thank the authors for their very thorough response which addressed all of my comments and clarified the areas I found unclear. I think the paper is now suitable for publication and will be an interesting and novel addition to the literature on the long-term outlook for Greenland's meltwater buffering capacity in firm.

Dear editor and reviewers, we would like to thank you for your final comments. We hope that our changes have improved the revised manuscript.

Reviewer #5 (Remarks to the Author):

I would like to thank the authors for their very thorough response which addressed all of my comments and clarified the areas I found unclear. I think the paper is now suitable for publication and will be an interesting and novel addition to the literature on the long-term outlook for Greenland's meltwater buffering capacity in firm.

Thank you very much for your thorough comments that have really helped us improving the manuscript.